# TOWARDS SYN-TO-REAL IQA: A NOVEL PERSPECTIVE ON RESHAPING SYNTHETIC DATA DISTRIBUTIONS

## ABSTRACT

Blind Image Quality Assessment (BIQA) has advanced significantly through deep learning, but the scarcity of large-scale labeled datasets remains a challenge. While synthetic data offers a promising solution, models trained on existing synthetic datasets often show limited generalization ability. In this work, we make a key observation that representations learned from synthetic datasets often exhibit a discrete and clustered pattern that hinders regression performance: features of high-quality images cluster around reference images, while those of low-quality images cluster based on distortion types. Our analysis reveals that this issue stems from the distribution of synthetic data rather than model architecture. Consequently, we introduce a novel framework SynDR-IQA, which reshapes synthetic data distribution to enhance BIQA generalization. Based on theoretical derivations of sample diversity and redundancy's impact on generalization error, SynDR-IQA employs two strategies: distribution-aware diverse content upsampling, which enhances visual diversity while preserving content distribution, and density-aware redundant cluster downsampling, which balances samples by reducing the density of densely clustered areas. Extensive experiments across three cross-dataset settings (synthetic-to-authentic, synthetic-to-algorithmic, and synthetic-to-synthetic) demonstrate the effectiveness of our method. Additionally, as a data-based approach, SynDR-IQA can be coupled with model-based methods without increasing inference costs. The source code will be publicly available.

## 1 INTRODUCTION

Blind Image Quality Assessment (BIQA) aims to automatically and accurately evaluate image quality without relying on reference images Li et al. (2016); Yan et al. (2019). It plays a crucial role in enhancing user experience in multimedia applications, improving the robustness of downstream image processing algorithms, and guiding the optimization of image enhancement methods. However, the BIQA task is challenging due to its complexity and high association with human perception.

In recent years, mainstream BIQA methods have greatly surpassed traditional methods owing to the powerful representational capabilities of deep learning models. However, the success of deep learning largely relies on large-scale annotated datasets. The high cost of acquiring subjective quality labels limits the growth of existing datasets. The availability of reference images and the controllability of quality degradation in synthetic distortions suggest that low-cost data augmentation through artificially synthesized data appears to be a feasible solution. In practice, training directly on existing synthetic distortion datasets results in suboptimal quality representations with limited generalization capabilities.

We observe a key phenomenon: models trained on synthetic data tend to produce a discrete and clustered feature distribution. Specifically, as shown in Fig. 1 high-quality image features form distinct clusters based on reference images, while low-quality image features gather based on distortion types. Medium-quality image features lack smooth transitions and tend to attach to high/low-quality clusters. This discontinuous representation is detrimental to the performance of regression tasks like BIQA Zha et al. (2022); Li et al. (2024b). We believe that this phenomenon is primarily caused by two core problems in synthetic distortion datasets:

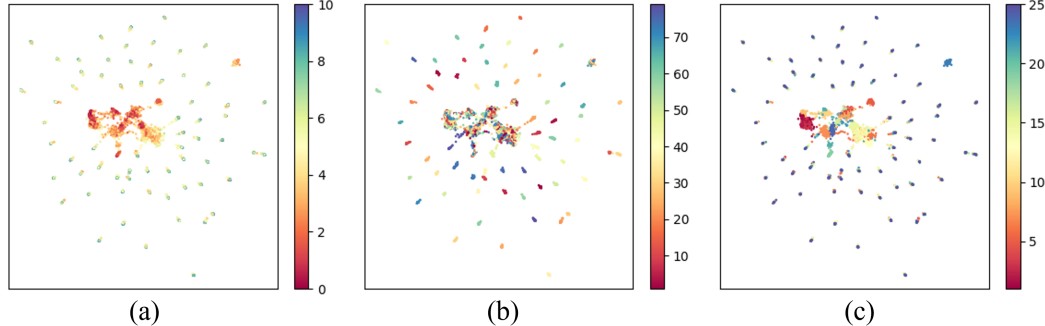

(a)                              (b)                              (c)

Figure 1: (a), (b), and (c) present UMAP McInnes et al. (2018) visualizations of the same representation learned from the synthetic distortion dataset KADID-10k Lin et al. (2019) by the baseline model He et al. (2016). The different colors in each figure convey different meanings: in (a), the colors represent the corresponding Mean Opinion Score (MOS) values, with higher values indicating better quality; in (b), the colors correspond to the reference images; and in (c), the colors denote the types of distortions.

- 1) **Insufficient content diversity**, which is caused by the limited reference images in synthetic distorted datasets. This leads to the model's tendency to overfit, hindering the formation of a globally consistent quality representation.

- 2) **Excessive redundant samples**, which stem from the distorted images in synthetic distorted datasets being uniform combinations of reference images, distortion types, and distortion intensities. This induces the model to overly focus on these repetitive patterns while neglecting broader information, thereby exacerbating the overfitting problem.

To understand these issues thoroughly, we theoretically derive the impact of sample diversity and redundancy on generalization error. Based on this theoretical foundation, we design a framework called SynDR-IQA from a novel perspective, which reshapes the synthetic data distribution to improve the generalization ability of BIQA. Specifically, **to address the issue of insufficient content diversity**, we propose a Distribution-aware Diverse Content Upsampling (DDCUp) strategy. By sampling reference images from an unlabeled candidate reference set based on the content distribution of existing training set to generate distorted images, we increase the diversity of visual instances, helping the model learn consistent representations across different content. To label the newly generated distorted images, we employ a key assumption: similar content under the same distortion conditions should result in similar quality degradation. Based on this assumption, we generate pseudo-labels for the newly generated images referencing given labeled data corresponding to similar reference images in the training set, ensuring the reasonableness and consistency of the generated pseudo-labels. **To address the issue of excessive redundant samples**, we design a Density-aware Redundant Cluster Downsampling (DRCDown) strategy. It identifies high-density redundant clusters in the training dataset and selectively removes samples from these clusters while retaining samples from low-density regions. This mitigates the negative impact of redundant samples while alleviating data distribution imbalance, thus helping the model learn more generalizable representations. Our contributions can be summarized as follows:

- We observed the key phenomenon that models trained on synthetic data exhibit discrete and clustered feature distributions, and provide an in-depth analysis of the underlying causes. Through theoretical derivation, we demonstrate the impact of sample diversity and redundancy on the model's generalization error.

- From a novel perspective of reshaping synthetic data distribution, we proposed the SynDR-IQA framework, which includes a DDCUp strategy and a DRCDown strategy, to enhance the generalization capability of BIQA models.

- Extensive experiments across various cross-dataset settings, including synthetic-to-authentic, synthetic-to-synthetic, and synthetic-to-algorithmic, validated the effectiveness of the SynDR-IQA framework. Additionally, as a data-based approach, SynDR-IQA can be integrated with existing model-based methods without adding inference costs.

## 2 RELATED WORK

**Deep Learning-based BIQA Methods.** Deep learning has revolutionized BIQA, leading to significant advancements in accuracy and robustness Kang et al. (2014); Kim & Lee (2016). Recent works have explored various innovative approaches to address the challenges in this field. Zhu et al. Zhu et al. (2020) proposed MetaIQA, employing meta-learning to enhance generalization across diverse distortion types. Su et al. Su et al. (2020) introduced a self-adaptive hyper network architecture for adaptive quality estimation in real-world scenarios. Ke et al. Ke et al. (2021) developed MUSIQ, a multi-scale image quality transformer processing native resolution images with varying sizes. Shin et al. Shin et al. (2024) proposed QCN, utilizing comparison transformers and score pivots for improving cross-dataset generalization. Xu et al. Xu et al. (2024) demonstrated the effectiveness of injecting local distortion features into large pretrained vision transformers for IQA tasks. Despite these advancements, the success of deep learning-based BIQA methods heavily relies on large-scale annotated datasets. The high cost and time-consuming nature of acquiring subjective quality labels for real-world images significantly limit the growth of existing datasets. This limitation has prompted researchers to explore the potential of leveraging synthetic distortions to generalize to real-world distortions.

**Synthetic-to-Real Generalization in BIQA.** Due to the significant domain differences between synthetic distortions and real-world distortions, models trained on synthetic distortion data often perform poorly when facing real-world images. To bridge this gap, several studies have explored Unsupervised Domain Adaptation (UDA) techniques. Chen et al. Chen et al. (2021b) proposed a curriculum-style UDA approach for video quality assessment, adapting models from source to target domains progressively. Lu et al. Lu et al. (2022) introduced StyleAM, aligning source and target domains in the feature style space, which is more closely associated with image quality. Li et al. Li et al. (2023) developed FreqAlign, which excavates perception-oriented transferability from a frequency perspective, selecting optimal frequency components for alignment. Most recently, Li et al. Li et al. (2024a) proposed DGQA, a distortion-guided UDA framework that leverages adaptive multi-domain selection to match data distributions between source and target domains, reducing negative transfer from outlier source domains. However, previous work has neglected the distributional issues of synthetic distortion datasets. In this work, we introduce a novel framework SynDR-IQA to enhance the syn-to-real generalization ability of BIQA by reshaping the distribution of synthetic data.

## 3 METHODOLOGY

In this section, we introduce the SynDR-IQA framework, which aims to enhance the generalization capability of BIQA models by reshaping the synthetic data distribution. We begin with a problem formalization for BIQA, establishing the foundational context for our work. Then, we conduct a theoretical analysis exploring the impact of sample diversity and redundancy on the generalization error. Building upon this theoretical foundation, we detail the two core components of SynDR-IQA: Distribution-aware Diverse Content Upsampling (DDCUp), which addresses the challenge of limited diversity in synthetic datasets and Density-aware Redundant Cluster Downsampling (DRC-Down), which mitigates the issue of data redundancy.

### 3.1 PROBLEM FORMALIZATION

BIQA aims to predict the perceptual quality of images without reference. Let $\mathcal{X}$ denote the space of all possible images, and $\mathcal{Y} = [0, 1]$ represent the range of quality scores (for simplicity). The BIQA task is formalized as learning a function $f : \mathcal{X} \rightarrow \mathcal{Y}$ that maps an input image to its quality score. The optimal function $f^*$ is defined by minimizing the expected risk: $f^* = \arg\min_{f \in \mathcal{F}} \mathbb{E}_{(x,y) \sim \mathcal{D}}[\mathcal{L}(f(x), y)]$, where $\mathcal{F}$ is the hypothesis space, $\mathcal{D}$ is the true distribution of images and quality scores, and $\mathcal{L}$ is a loss function. Since the true data distribution is inaccessible, in practice, we instead minimize the empirical error on the training dataset $\hat{\mathcal{D}} = \{(x_i, y_i)\}_{i=1}^n$: $\hat{f} = \arg\min_{f \in \mathcal{F}} \frac{1}{n} \sum_{i=1}^n \mathcal{L}(f(x_i), y_i)$.

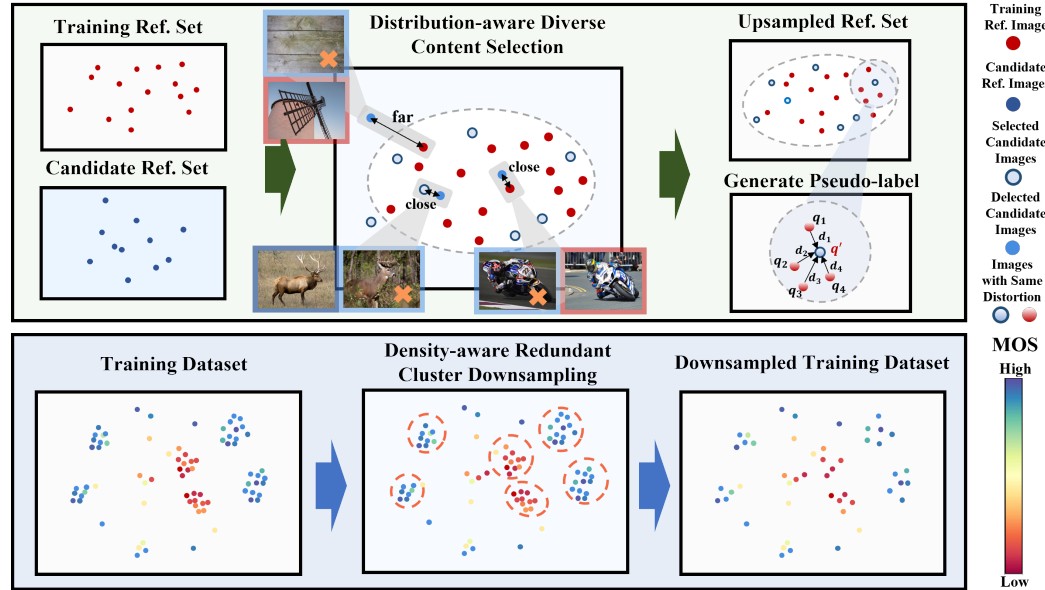

Figure 2: This figure shows the core concepts of two strategies in SynDR-IQA. The DDCUp strategy (upper) selects images from candidate reference sets that are similar in distribution but diverse in content to the training reference sets for synthesizing distorted samples. The pseudo-labels of these samples depend on the nearest neighbors of their reference images. The DRCDown strategy (lower) identifies high-density clusters in the training dataset and selectively removes samples from these clusters while retaining samples from low-density regions.

## 3.2 THEORETICAL ANALYSIS

The construction of synthetic distortion datasets follows a systematic process: applying predefined distortion types at various intensity levels to a set of reference images Lin et al. (2019). This generation mechanism exhibits two key characteristics: First, low-intensity distortions produce images nearly identical to their references, while high-intensity distortions generate images predominantly characterized by distortion-specific patterns. Second, the dataset generation typically employs uniform sampling across reference images, distortion types, and intensity levels. These two characteristics jointly lead to a fundamental issue: samples are drawn from different local distributions rather than following identical sampling from the true distribution, resulting in a discretely clustered structure in the distribution space.

To understand how this clustered distribution affects model generalization, we need to extend the classical generalization error analysis to account for samples being drawn from different local distributions rather than the true underlying distribution. To formalize this extension, we model the synthetic dataset $\hat{\mathcal{D}} = \{(x_i, y_i)\}_{i=1}^n$ as comprising $m$ clusters, where each cluster consists of one i.i.d. sample from the true distribution $\mathcal{D}$, along with its associated $k_i - 1$ samples drawn independently from the corresponding local distribution $\mathcal{D}_i \subset \mathcal{D}$. This formal characterization leads to the following generalization bound:

**Theorem 1** (Generalization Bound for Clustered Synthetic Data). *Let $\mathcal{F}$ be a hypothesis class of functions $f : \mathcal{X} \to \mathcal{Y}$, and $\hat{\mathcal{D}} = \{(x_i, y_i)\}_{i=1}^n$ be a dataset consisting of $m$ i.i.d. samples from true distribution $\mathcal{D}$, along with their respective $k_i - 1$ redundant samples independently drawn from the corresponding local distributions $\mathcal{D}_i \subset \mathcal{D}$. With probability at least $1 - \delta$, for all $f \in \mathcal{F}$, we have:*

$$|R(f) - R_{emp}(f)| \leq 2\operatorname{Rad}_m(\mathcal{F}) + \sqrt{\frac{2\log(2/\delta)}{n}} + \sqrt{\frac{\eta\log(2/\delta)}{8m}} + \frac{2\log(2/\delta)}{3m} \quad (1)$$

*where $R(f)$ is the true risk, $R_{emp}(f)$ is the empirical risk, $\mathrm{Rad}_m(\mathcal{F})$ is the Rademacher complexity based on $m$ distinct samples, $n = \sum_{i=1}^{m} k_i$ is the total number of samples, and $\eta = \frac{1}{m}\sum_{i=1}^{m}\frac{1}{k_i}$ is defined as redundancy heterogeneity which quantifies the degree of imbalanced distribution of redundant samples in the dataset.*

*Proof.* Please refer to the Appendix A.1 for complete proof. $\square$

According to Theorem 1, we can observe that the upper bound of the generalization error is influenced by the Rademacher complexity $\mathrm{Rad}_m(\mathcal{F})$, total sample size $n$, number of *i.i.d.* samples $m$ (also referred to as diverse samples), and redundancy heterogeneity $\eta$. Excluding model-related factors (captured by $\mathrm{Rad}_m(\mathcal{F})$), the bound reveals that the number of distinct samples $m$ plays a primary role in determining the upper bound. Enhancing the sampling of diverse samples can effectively lower this upper bound. Balancing samples to reduce redundancy heterogeneity $\eta$ can also effectively decrease the upper bound of the generalization error. While increasing redundant samples from local distributions can enlarge $n$ and reduce the second term, it may lead to higher redundancy heterogeneity $\eta$, potentially amplifying the third term degrading overall generalization performance.

These theoretical insights motivate us to reshape the sample distribution from the perspectives of **increasing content diversity** (increasing $m$) and **balancing sample density** (decreasing $\eta$) to improve the generalization performance of BIQA.

### 3.3 SYNDR-IQA FRAMEWORK

Building upon the theoretical insights, we propose the SynDR-IQA framework as shown in Fig. 3, which consists of two primary strategies: distribution-aware diverse content upsampling and density-aware redundant cluster downsampling. These strategies collectively aim to reshape the synthetic data distribution to obtain more generalizable BIQA models.

### 3.4 DISTRIBUTION-AWARE DIVERSE CONTENT UPSAMPLING

To enhance the generalization ability of BIQA models, we introduce a DDCUp strategy. This method aims to enrich the training data with diverse visual content while preserving the underlying content distribution of the original training set.

---

**Algorithm 1** Distribution-aware Diverse Content Upsampling Strategy

---

**Input:** Training dataset $\mathcal{D}$, Training reference set $\mathcal{D}_r$, Candidate reference set $\mathcal{D}_c$, Feature extractor $f(\cdot)$, Distance metric $\mathrm{Dist}(\cdot)$
**Output:** Upsampled training dataset $\mathcal{D}'$
1: Initialize $\mathcal{D}_r^{new} \leftarrow \emptyset$
2: $DistT \leftarrow \{\mathrm{Dist}(f(x_r^i), f(x_r^j)) | x_r^i, x_r^j \in \mathcal{D}_r, i \neq j\}$
3: **for** each $x_c \in \mathcal{D}_c$ **do**
4: $\quad DistC \leftarrow \mathrm{Dist}(f(x_c), f(\mathcal{D}_r))$
5: $\quad$ **if** $\mathrm{Min}(DistC) > \mathrm{Median}(DistT)$ and $\mathrm{Max}(DistC) < \mathrm{Max}(DistT)$ **then**
6: $\quad\quad$ **if** $\mathcal{D}_r^{new}$ is $\{\}$ **then**
7: $\quad\quad\quad \mathcal{D}_r^{new} \leftarrow \mathcal{D}_r^{new} \cup \{x_c\}$
8: $\quad\quad$ **else**
9: $\quad\quad\quad DistN \leftarrow \mathrm{Dist}(f(x_c), f(\mathcal{D}_r^{new}))$
10: $\quad\quad\quad$ **if** $\mathrm{Min}(DistN) > \mathrm{Median}(DistT)$ **then**
11: $\quad\quad\quad\quad \mathcal{D}_r^{new} \leftarrow \mathcal{D}_r^{new} \cup \{x_c\}$
12: $\quad\quad\quad$ **end if**
13: $\quad\quad$ **end if**
14: $\quad$ **end if**
15: **end for**
16: $\mathcal{D}' \leftarrow \mathcal{D} \cup \{\mathcal{D}, \mathcal{D}_r, \mathrm{GenSyn}(\mathcal{D}_r^{new}, ...)\}$
17: **return** $\mathcal{D}'$

---

Algorithm 1 details the DDCUp strategy. We select additional reference images from KADIS-700k Lin et al. (2019) to build a candidate reference set. To avoid introducing excessive noise, we limit its size to be equal to that of training reference set. A feature extractor $f$, pre-trained on ImageNet, is used to extract features of the reference images from both the training reference set and the candidate reference set. A distance metric, $\text{Dist}(\cdot)$ (cosine distance in our implementation), is utilized to guide the selection process. Our algorithm ensures that the selected images are similar in content to the training dataset while also being sufficiently distinct from each other (lines 5 and 10), thereby promoting diversity and preventing redundancy.

---

**Algorithm 2** Synthetic Data Generation

---

**Input:** Training dataset $\mathcal{D}$, Training reference set $\mathcal{D}_r$, Selected referance images from candidate reference set $\mathcal{D}_r^{new}$, Feature extractor $f(\cdot)$, Distance metric $\text{Dist}(\cdot)$, Number of nearest neighbors $k$, Distortion algorithm $\text{GenDist}(\cdot)$, Number of distortion types $T$, Number of distortion level $L$

**Output:** Generated synthetic dataset $\mathcal{D}_{GenSyn}$

1: Initialize $\mathcal{D}_{GenSyn} \leftarrow \emptyset$
2: **for** each $x_{new} \in \mathcal{D}_r^{new}$ **do**
3:    $nns \leftarrow \text{kNN}(f(x_{new}), f(\mathcal{D}_r), k, \text{Dist})$ {k-Nearest Neighbors of $x_{new}$}
4:    $nns \leftarrow \{nn \mid \text{Dist}(f(x_r^{nns[0]}), f(x_{new})) - \text{Dist}(f(x_r^{nn}), f(x_{new})) < 0.05, nn \in nns\}$
5:    $w \leftarrow \text{Softmax}(\{\text{Dist}(f(x_r^{nn}), f(x_{new})) \mid nn \in nns\})$
6:    **for** each $t \in T$ **do**
7:      **for** each $l \in L$ **do**
8:        $x_{new}^{(t,l)} \leftarrow \text{GenDist}(x_{new}, t, l)$
9:        $y_{new}^{(t,l)} \leftarrow \sum_{nn \in nns} w_{nn} y_{nn}^{(t,l)}$
10:        $\mathcal{D}_{GenSyn} \leftarrow \mathcal{D}_{GenSyn} \cup \{(x_{new}^{(t,l)}, y_{new}^{(t,l)})\}$
11:      **end for**
12:    **end for**
13: **end for**
14: **return** $\mathcal{D}_{GenSyn}$

---

After selecting diverse reference images, we generate corresponding distorted images and pseudo-labels to augment the training dataset. Algorithm 2 details this process. For each selected reference image, we apply the same distortion generation process used to create the training dataset. Notably, to reduce the increase of redundant samples and prevent additional label noise, only distortion intensities of levels 1, 3, and 5 are implemented. To generate reliable pseudo-labels for these newly distorted images, we leverage the assumption that similar content under the same distortion conditions should result in similar quality degradation. For each new reference image $x_{new}$, we identify its k nearest neighbors (kNN) within the original training set's reference images based on the features' distances. To further enhance the reliability of the pseudo-labels, we filter these nearest neighbors, retaining only those whose feature distance to $x_{new}$ is within a certain threshold (line 4). The pseudo-label for each distorted image of $x_{new}$ is then calculated as a weighted average of the labels of its nearest neighbors' corresponding distorted images, where the weights are determined by the softmax of the features' distances.

### 3.5 DENSITY-AWARE REDUNDANT CLUSTER ROWNSAMPLING

An overabundance of redundant samples can bias the model, hindering its ability to generalize to unseen data. It also contributes to increased redundancy heterogeneity ($\eta$ in Theorem 1), further increasing the generalization error. Therefore, we propose a DRCDown strategy to mitigate the negative impact of redundant samples and reduce $\eta$, thereby further enhancing the model's generalization performance.

Algorithm 3 details the DRCDown strategy. Similar to the DDCUp strategy, we utilize the feature extractor $f(\cdot)$ to obtain feature representations of the training samples. We then identify pairs of similar samples based on both the distance metric $\text{Dist}(\cdot)$ and label distance (L1 in our implementation) (line 5). It ensures that we remove redundancy without discarding hard samples with tiny feature difference yet large quality difference. The thresholds for the distances of feature and label are empirically set to 0.1, and 1 (for MOS values in $[0, 10]$), separately. By considering both feature

and label similarity, we aim to specifically target and reduce the density of high-density clusters that contribute significantly to redundancy heterogeneity.

---

**Algorithm 3** Density-aware Redundant Cluster Downsampling Strategy

---

**Input:** Training dataset $\mathcal{D}$, Feature extractor $f(\cdot)$, Distance metric $Dist(\cdot)$, Number of samples $N$, Threshold for minimum union size $T_u$
**Output:** Downsampled dataset $\mathcal{D}_{down}$
 1: Initialize $\mathcal{D}_{down} \leftarrow \emptyset$
 2: Initialize $SimPairs \leftarrow \emptyset$
 3: **for** $i \leftarrow 1$ to $N - 1$ **do**
 4:    **for** $j \leftarrow i + 1$ to $N$ **do**
 5:      **if** $Dist(f(x_i), f(x_j)) < 0.1$ **and** $|y_i - y_j| < 1$ **then**
 6:        $SimPairs \leftarrow SimPairs \cup \{(x_i, x_j)\}$
 7:      **end if**
 8:    **end for**
 9: **end for**
10: $Unions \leftarrow \text{DSU}(SimPairs)$ {Union disjoint sets of similar pairs}
11: **for** each $u \in Unions$ **do**
12:    **if** $\text{Length}(u)/2 > T_u$ **then**
13:      $u \leftarrow \{$randomly select $\text{Max}(\lfloor \text{Length}(u)/2 \rfloor, T_u)$ samples among union$\}$
14:    **end if**
15:    $\mathcal{D}_{down} \leftarrow \mathcal{D}_{down} \cup u$
16: **end for**
17: **return** $\mathcal{D}_{down}$

---

After identifying similar sample pairs, we employ a disjoint set union (DSU) data structure to group these pairs into clusters (line 9). For each cluster whose size is greater than $2T_u$, we randomly downsample it to $\max(\lfloor N_u/2 \rfloor, T_u)$ samples, where $N_u$ is the original cluster size (line 12). This threshold $T_u$ prevents excessive downsampling, ensuring that the downsampled dataset retains sufficient information for effective training. By selectively removing samples from over-represented regions, the DRCDown strategy effectively reduces redundancy and promotes a more balanced data distribution, directly addressing the issue of high redundancy heterogeneity and thereby contributing to improved generalization performance. This reduction in $\eta$ helps to lower the generalization error bound as established in Theorem 1, leading to a more robust and generalizable IQA model.

## 4 Experiments

### 4.1 Experimental Setups

**Datasets and Protocols.** We conduct experiments on eight IQA datasets including four synthetic distortion datasets LIVE Sheikh et al. (2006), CSIQ Chandler (2010), TID2013 Ponomarenko et al. (2013), KADID-10k Lin et al. (2019), three authentic distortion datasets LIVEC Ghadiyaram & Bovik (2015), KonIQ-10k Hosu et al. (2020), BID Ciancio et al. (2010), and the dataset PIPAL Jinjin Gu (2020) with both synthetic and algorithmic distortions. The models' performance in prediction accuracy and monotonicity is assessed using Spearman's Rank Correlation Coefficient (SRCC) and Pearson's Linear Correlation Coefficient (PLCC). Both coefficients range from -1 to 1, with values near 1 signifying better performance.

**Implementation Details.** In our experiments, we use the same model architecture (ResNet-50) and loss function (L1Loss) as DGQA Li et al. (2024a). For the synthetic-to-authentic and synthetic-to-algorithmic settings, we train the model using distortion types selected by DGQA, while in the synthetic-to-synthetic setting, training covers all distortion types. To prevent content overlap when training within datasets, we apply 80/20 split based on reference images. This random train-test splitting is repeated ten times, with the median SRCC and PLCC reported. For cross-database experiments, models are trained on KADID-10k and tested on other datasets. During training, we randomly sample one patch of resolution $224 \times 224$ from each image, and random horizontal flipping is employed for data augmentation. The mini-batch size is set to 32, with a learning rate of $2 \times 10^{-5}$. The Adam optimizer, with a weight decay of $5 \times 10^{-4}$, is used to optimize the model for 24

Table 1: Performance comparison on the synthetic-to-authentic setting (KADID-10k→LIVEC, KonIQ-10k, and BID). The average results are in the last column. And the bolded results imply top performance.

| Methods | LIVEC | | KonIQ-10k | | BID | | Average | |
|---|---|---|---|---|---|---|---|---|
| | SRCC | PLCC | SRCC | PLCC | SRCC | PLCC | SRCC | PLCC |
| RankIQA Liu et al. (2017) | 0.491 | 0.495 | 0.603 | 0.551 | 0.510 | 0.367 | 0.535 | 0.471 |
| DBCNN Zhang et al. (2018) | 0.572 | 0.589 | 0.639 | 0.618 | 0.620 | 0.609 | 0.613 | 0.606 |
| MetaIQA Zhu et al. (2020) | 0.464 | 0.464 | 0.501 | 0.504 | 0.301 | 0.428 | 0.422 | 0.465 |
| HyperIQA Su et al. (2020) | 0.490 | 0.487 | 0.545 | 0.556 | 0.379 | 0.282 | 0.472 | 0.442 |
| MUSIQ Ke et al. (2021) | 0.517 | 0.524 | 0.554 | 0.573 | 0.575 | 0.600 | 0.549 | 0.566 |
| GraphIQA Sun et al. (2022) | 0.388 | 0.407 | 0.427 | 0.430 | 0.524 | 0.533 | 0.446 | 0.456 |
| VCRNet Pan et al. (2022) | 0.561 | 0.548 | 0.517 | 0.525 | 0.542 | 0.545 | 0.540 | 0.540 |
| KGANet Zhou et al. (2024) | 0.575 | - | 0.528 | - | - | - | - | - |
| CLIPIQA Wang et al. (2023) | 0.643 | 0.629 | 0.684 | 0.702 | 0.627 | 0.581 | 0.651 | 0.637 |
| CLIPIQA+ Wang et al. (2023) | 0.512 | 0.543 | 0.511 | 0.515 | 0.474 | 0.442 | 0.499 | 0.500 |
| Q-Align Wu et al. (2024) | 0.702 | **0.744** | 0.668 | 0.665 | - | - | - | - |
| DANN Ajakan et al. (2014) | 0.499 | 0.484 | 0.638 | 0.636 | 0.586 | 0.510 | 0.574 | 0.543 |
| UCDA Chen et al. (2021b) | 0.382 | 0.358 | 0.496 | 0.501 | 0.348 | 0.391 | 0.408 | 0.417 |
| RankDA Chen et al. (2021a) | 0.451 | 0.455 | 0.638 | 0.623 | 0.535 | 0.582 | 0.542 | 0.553 |
| StyleAM Lu et al. (2022) | 0.584 | 0.561 | 0.700 | 0.673 | 0.637 | 0.567 | 0.640 | 0.600 |
| FreqAlign Li et al. (2023) | 0.618 | 0.588 | **0.748** | 0.721 | 0.674 | 0.708 | 0.680 | 0.673 |
| DGQA Li et al. (2024a) | 0.696 | 0.690 | 0.681 | 0.687 | 0.770 | 0.753 | 0.716 | 0.710 |
| SynDR-IQA | **0.713** | 0.714 | 0.727 | **0.735** | **0.788** | **0.764** | **0.743** | **0.737** |

epochs. In testing, five patches per image are sampled, and their average prediction is used as the final output. This implementation is carried out using PyTorch, with all experiments conducted on NVIDIA TITAN Xp GPUs.

## 4.2 PERFORMANCE EVALUATION

**1) Performance on the synthetic-to-authentic setting.** We first evaluate the generalization capability of SynDR-IQA when transferring from synthetic to authentic distortions. Specifically, we train the models on KADID-10k and test them on LIVEC, KonIQ-10k, and BID, respectively. The results of the comparison between SynDR-IQA and 17 classical or state-of-the-art BIQA methods, including 11 deep learning-based methods (RankIQA Liu et al. (2017), DB-CNN Zhang et al. (2018), MetaIQA Zhu et al. (2020), HyperIQA Su et al. (2020), MUSIQ Ke et al. (2021), GraphIQA Sun et al. (2022), VCRNet Pan et al. (2022), KGANet Zhou et al. (2024), CLIPIQA Wang et al. (2023), CLIPIQA+ Wang et al. (2023), Q-Align Wu et al. (2024)) and 6 UDA-based methods (DANN Ajakan et al. (2014), UCDA Chen et al. (2021b), RankDA Chen et al. (2021a), StyleAM Lu et al. (2022), FreqAlign Li et al. (2023), and DGQA Li et al. (2024a)), are summarized in Table 1.

In Table 1, it can be seen that SynDR-IQA achieves state-of-the-art performance across all three authentic datasets. Our method is only slightly outperformed by FreqAlign in KADID-10k→KonIQ-10k scenario (SRCC) and by Q-Align in KADID-10k→LIVEC scenario (PLCC). Compared to the next best method, DGQA, our method improves the average SRCC and PLCC across the three datasets by 2.7% and 2.7%, respectively. The significant improvement demonstrates the superior generalization capability of SynDR-IQA when transferring from synthetic to authentic distortions.

**2) Performance on the synthetic-to-algorithmic setting.** We further evaluate SynDR-IQA in the synthetic-to-algorithmic setting, where models trained on KADID-10k are tested on the algorithmic distortions in the PIPAL dataset. Table 2 compares our method with the baseline DGQA across different types of algorithmic distortions.

As shown in Table 2, SynDR-IQA achieves consistent improvements over DGQA for most distortion types. Notably, for PSNR-originated SR and GAN-based SR distortions, our method outperforms DGQA by over 4% in SRCC. The only exception is the Denoising category, where SynDR-IQA shows a slight decrease. However, the overall average performance indicates that our method effec-

Table 2: Performance comparison on the setting of synthetic-to-algorithmic (KADID-10k→algorithmic distortions on PIPAL). The average results are in the last row. The relative improvements over DGQA are indicated in subscript.

| Distortion Type | DGQA | | SynDR-IQA | |
|---|---|---|---|---|
| | SRCC | PLCC | SRCC | PLCC |
| Traditional SR | 0.5419 | 0.5404 | $\mathbf{0.5845}_{+4.26\%}$ | $\mathbf{0.5680}_{+2.76\%}$ |
| PSNR-originated SR | 0.5810 | 0.5956 | $\mathbf{0.6263}_{+4.53\%}$ | $\mathbf{0.6342}_{+3.86\%}$ |
| SR with kernel mismatch | 0.1629 | 0.1353 | $\mathbf{0.1998}_{+3.69\%}$ | $\mathbf{0.1629}_{+2.76\%}$ |
| GAN-based SR | 0.5393 | 0.5279 | $\mathbf{0.5749}_{+3.56\%}$ | $\mathbf{0.5552}_{+2.73\%}$ |
| Denoising | **0.5588** | **0.5193** | $0.5557_{-0.31\%}$ | $0.5023_{-1.70\%}$ |
| SR and Denoising Joint | 0.4470 | 0.4390 | $\mathbf{0.4835}_{+3.65\%}$ | $\mathbf{0.4691}_{+3.01\%}$ |
| Average | 0.4718 | 0.4596 | $\mathbf{0.5041}_{+3.23\%}$ | $\mathbf{0.4820}_{+2.24\%}$ |

Table 3: Performance comparison on the synthetic-to-synthetic setting (single database evaluation on KADID-10k, KADID-10k→LIVE, CSIQ, and TID2013). The average results are in the last row.

| Methods | KADID-10k | | LIVE | | CSIQ | | TID2013 | |
|---|---|---|---|---|---|---|---|---|
| | SRCC | PLCC | SRCC | PLCC | SRCC | PLCC | SRCC | PLCC |
| Baseline | 0.8528 | 0.8526 | 0.9173 | 0.8988 | 0.7965 | 0.8017 | 0.7077 | 0.7220 |
| SynDR-IQA | **0.8922** | **0.8974** | **0.9258** | **0.9014** | **0.8069** | **0.8092** | **0.7147** | **0.7328** |

tively generalizes to algorithmic distortions, demonstrating its robustness in handling complex and unseen distortion types.

**3) Performance on the synthetic-to-synthetic setting.** To evaluate the effectiveness of SynDR-IQA within synthetic distortion scenarios, we conduct both single-dataset evaluation on KADID-10k and cross-dataset evaluations, training on KADID-10k and testing on LIVE, CSIQ, and TID2013. The results are summarized in Table 3.

From Table 3, SynDR-IQA demonstrates superior performance over the baseline in both in-dataset and cross-dataset evaluations. On KADID-10k, our method achieves 0.8922 (SRCC), indicating better fitting to the training data. In cross-dataset testing, SynDR-IQA maintains higher performance, showing enhanced generalization to other synthetic distortion datasets. These results confirm that our approach enables the model to learn more robust and generalized feature representations.

### 4.3    ABLATION STUDY

To understand the contributions of each component in SynDR-IQA, we perform a series of ablation experiments on the synthetic-to-authentic setting, as shown in Table 4. The results are reported in terms of SRCC. In this table, *CD* refers to adding the full candidate dataset in training, *CD+SEL* denotes DDCUp, and *DOWN* signifies DRCDown.

**Effect of Candidate Dataset.** Comparing **a)** and **b)**, we observe that while the performance on KonIQ-10k improves, there is a slight decrease in performance on LIVEC and BID. This suggests that simply increasing data content is not always beneficial. Introducing the complete candidate dataset enriches content diversity but also includes samples with significant distributional differences from the training dataset and new redundant samples, which may hinder model generalization.

**Effect of Distribution-aware Diverse Content Upsampling.** Comparing **a)**, **b)**, and **e)**, we find that by selectively increasing diversity while maintaining data distribution through our DDCUp strategy, the model's performance improves significantly. This approach effectively balances content diversity and distribution consistency, leading to better generalization across different datasets.

**Effect of Density-aware Redundant Cluster Downsampling.** Comparing **a)** and **c)**, we note that implementing DRCDown alone already shows a consistent improvement over the baseline. This indicates that controlling sample density effectively addresses data redundancy and imbalance issues, resulting in more precise and generalizable feature representations.

Table 4: Ablation study of SynDR-IQA components on the synthetic-to-authentic setting. The symbols ✓ and indicate the inclusion of a component.

| Index | CD | SEL | DOWN | LIVEC | KonIQ-10k | BID | Average |
|---|---|---|---|---|---|---|---|
| **a)** | | | | 0.6958 | 0.6810 | 0.7696 | 0.7155 |
| **b)** | ✓ | | | 0.6901 | 0.7105 | 0.7677 | 0.7228 |
| **c)** | | | ✓ | 0.6962 | 0.6887 | 0.7793 | 0.7214 |
| **d)** | ✓ | | ✓ | 0.7095 | 0.7107 | 0.7775 | 0.7326 |
| **e)** | ✓ | ✓ | | **0.7219** | 0.7119 | 0.7822 | 0.7387 |
| **f)** | ✓ | ✓ | ✓ | $0.7127_{+1.69\%}$ | $\mathbf{0.7268}_{+4.58\%}$ | $\mathbf{0.7884}_{+1.88\%}$ | $\mathbf{0.7426}_{+2.71\%}$ |

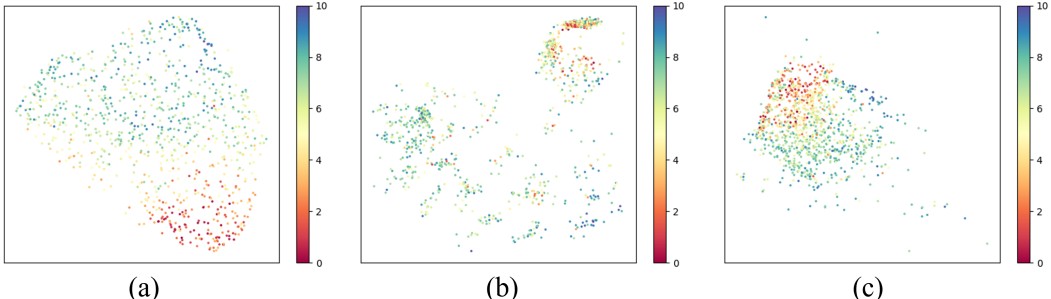

|  (a)  |  (b)  |  (c)  |

Figure 3: (a), (b), and (c) show the UMAP visualization of features extracted from LIVEC using the same model under different training processes: (a) is trained directly within the LIVEC; (b) is trained on KADID-10k based on DGQA; (c) is trained on KADID-10k based on SynDR-IQA.

The combination of all components yields the best overall performance, achieving a remarkable 2.71% improvement in average SRCC compared to the baseline. These results demonstrate the impact of the components we proposed in SynDR-IQA, which together contribute to a more robust and generalized model.

**Visualization Analysis.** We implemented the same model under three different training processes: directly on the LIVEC database, on KADID-10k based on DGQA, and on KADID-10k based on SynDR-IQA. Using the trained models, we extracted features from images in LIVEC for UMAP visualization, as shown in Fig. 4.3. It can be observed that the quality representation obtained from training on KADID-10k based on DGQA remains relatively dispersed. However, by using SynDR-IQA, we achieve representations closer to those obtained from training on authentic distortion samples, thus validating the effectiveness of our method.

## 5 CONCLUSION

In this work, we aim to address the critical challenge of limited generalization ability in BIQA models trained on synthetic datasets. Our investigation reveals a key pattern: representations learned from synthetic datasets often exhibit discrete and clustered distributions, with high-quality image features clustering around reference images and low-quality features clustering based on distortion types, which significantly hinders regression performance in BIQA tasks. Guided by this observation, we conduct theoretical derivations to understand the impact of sample diversity and redundancy on generalization error. This theoretical foundation leads to the development of our novel framework, SynDR-IQA, designed to reshape synthetic data distribution for enhancing BIQA generalization. The proposed SynDR-IQA consists of two key strategies: 1) DDCUp, which enhances content diversity while preserving the content distribution of the training dataset; 2) DRCDown, which optimizes sample distribution by reducing the density of dense clusters. We validate the effectiveness of SynDR-IQA through extensive experiments across three cross-dataset settings (synthetic-to-authentic, synthetic-to-algorithmic, and synthetic-to-synthetic). The results consistently demonstrate improved generalization ability of BIQA models trained with our framework.

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

# A APPENDIX

## A.1 THE PROOF OF THEOREM 1

### Step 1. Building Generalization Error.

We define the supremum of the absolute difference between the true risk $R(f)$ and the empirical risk $R_{\text{emp}}(f)$:

$$\Phi(X) = \sup_{f \in \mathcal{F}} |R(f) - R_{\text{emp}}(f)|.$$

By applying McDiarmid's inequality, with probability at least $1 - \delta/2$:

$$\Phi(X) \leq \mathbb{E}[\Phi(X)] + \sqrt{\frac{2 \log(2/\delta)}{n}},$$

where $n$ is the total number of samples.

### Step 2. Bounding with Rademacher Complexity.

We upper bound the expectation using Rademacher complexity:

$$\mathbb{E}[\Phi(X)] \leq 2 \, \text{Rad}_m(\mathcal{F}),$$

where $\text{Rad}_m(\mathcal{F})$ is the empirical Rademacher complexity based on $m$ *i.i.d* samples.

### Step 3. Bounding Additional Error from Clustered Data Distribution.

**Dataset Decomposition.** To simplify the analysis, we assume that redundant samples drawn from the corresponding local distributions $\mathcal{D}_i \subset \mathcal{D}$ have the same input $x$ but potentially different labels $y$. Since samples are independent, considering different inputs $x$ does not affect the generality of the conclusion.

We decompose the dataset into $m$ *i.i.d* inputs $\{x_1, x_2, \ldots, x_m\}$, where each input $x_i$ is independently repeated $k_i$ times with possibly different labels $y_{i1}, y_{i2}, \ldots, y_{ik_i}$. For each input $x_i$, define the random variable $Y_i$ representing the average loss over its repetitions:

$$Y_i = \frac{1}{k_i} \sum_{j=1}^{k_i} l(f(x_i), y_{ij}),$$

where $l(f(x_i), y_{ij})$ denotes the loss function evaluated at $x_i$ with label $y_{ij}$.

**Bounding Variance of $Y_i$.** Since $\mathcal{Y} = [0, 1]$ and the loss function is L1, we have $l(f(x_i), y_{ij}) \in [0, 1]$ and can upper bound the variance of $Y_i$:

$$\text{Var}(Y_i) \leq \frac{1}{4k_i}.$$

Figure 4: Qualitative results of SynDR-IQA on LIVEC. For each image, the top number represents the human-annotated ground-truth score, normalized to the range [0, 10]; the middle number represents our model's predicted quality score; and the bottom number represents the quality score predicted by CLIP-IQA Wang et al. (2023). The ground-truth scores of these images progressively increase from left to right and from top to bottom.

**Application of Bernstein's Inequality.** We define

$$Z = \frac{1}{m} \sum_{i=1}^{m} (Y_i - \mu_i),$$

where $\mu_i = \mathbb{E}[Y_i]$.

Applying Bernstein's inequality, with probability at least $1 - \delta/2$:

$$|Z| \leq \sqrt{\frac{\eta \log(2/\delta)}{8m}} + \frac{2 \log(2/\delta)}{3m},$$

where $\eta = \frac{1}{m} \sum_{i=1}^{m} \frac{1}{k_i}$.

**4. Final Generalization Error Bound.** Combining the results from steps 1, 2 and 3, and using the union bound to ensure a total probability of at least $1 - \delta$, we finally obtain the generalization error bound:

$$|R(f) - R_{\text{emp}}(f)| \leq 2 \operatorname{Rad}_m(\mathcal{F}) + \sqrt{\frac{2 \log(2/\delta)}{n}} + \sqrt{\frac{\eta \log(2/\delta)}{8m}} + \frac{2 \log(2/\delta)}{3m}.$$

## A.2 QUALITATIVE RESULTS

To qualitatively demonstrate the effectiveness of our method, we showcase several representative examples from LIVEC in Fig. A.1. The examples span diverse scenarios with various quality scores, distortions, scenes, and content. Notably, our model, trained solely on the synthetic distortion dataset KADID-10k, generates predictions that align well with human-annotated ground truth scores on these real-world images, validating the effective synthetic-to-real generalization capability of our

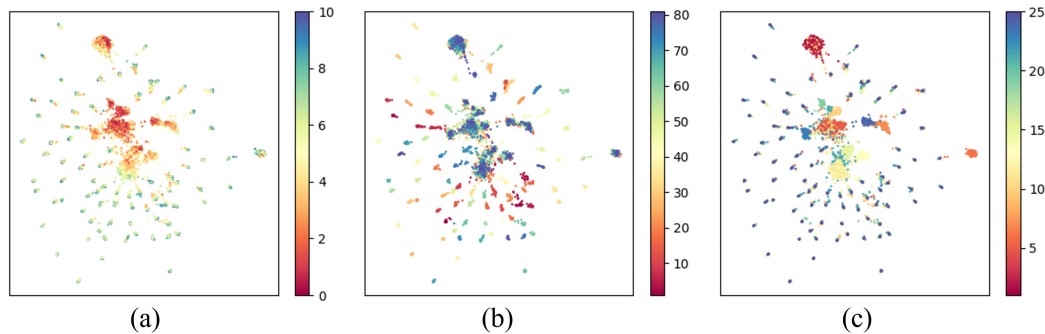

(a)                 (b)                 (c)

Figure 5: (a), (b), and (c) present UMAP McInnes et al. (2018) visualizations of the same representation learned from the synthetic distortion dataset KADID-10k Lin et al. (2019) by Swin-T Liu et al. (2021). The different colors in each figure convey different meanings: in (a), the colors represent the corresponding Mean Opinion Score (MOS) values, with higher values indicating better quality; in (b), the colors correspond to the reference images; and in (c), the colors denote the types of distortions.

Table 5: Performance comparison between baseline and SynDR-IQA using Swin-T backbone on the synthetic-to-authentic setting.

| Method | LIVEC | | KonIQ-10k | | BID | | Average | |
|---|---|---|---|---|---|---|---|---|
| | SRCC | PLCC | SRCC | PLCC | SRCC | PLCC | SRCC | PLCC |
| Baseline | 0.620 | 0.641 | 0.600 | 0.625 | 0.729 | 0.684 | 0.649 | 0.650 |
| SynDR-IQA | **0.670** | **0.693** | **0.719** | **0.752** | **0.777** | **0.750** | **0.721** | **0.731** |

approach. Furthermore, compared to the state-of-the-art method CLIP-IQA Wang et al. (2023), our approach shows significantly better alignment with human perception.

### A.3 Architecture-agnostic Analysis

To demonstrate that our findings and method are architecture-agnostic, we conduct additional experiments using Swin Transformer Tiny (Swin-T) Liu et al. (2021) as the backbone network, following the same experimental protocol as our main experiments.

The UMAP visualization of features learned by Swin-T on KADID-10k (Fig. A.3) exhibits patterns consistent with our previous observations, confirming that the identified dataset characteristics, rather than architectural choices, are the primary source of the generalization challenges.

As shown in Table 5, our method significantly improves the baseline performance with Swin-T backbone, achieving an average SRCC improvement of 7.2%. These results support two key claims: 1) the generalization challenges stem from dataset properties rather than architectural limitations, and 2) our proposed SynDR-IQA framework is effective across different neural architectures, demonstrating the generality of our approach.

### A.4 Comparison with Models Trained on Authentic Distortion Dataset

To further demonstrate the effectiveness of our SynDR-IQA framework, we compare our model trained only on synthetic data (KADID-10k) with state-of-the-art models trained on SPAQ Fang et al. (2020), which is a large-scale authentic distortion dataset containing 11125 images. The comparative experimental results are from paper Wu et al. (2024). This comparison aims to investigate how well our synthetic-trained model can approach the performance of models trained on authentic distortions.

Table 6 shows the performance comparison on LIVEC and KonIQ-10k. Despite using only synthetic training data, our method achieves competitive performance compared to models trained on authentic distortions, with only a minor performance gap.

Table 6: Performance comparison with SOTA models trained on authentic distortion dataset SPAQ.

| Methods | Training Dataset | LIVEC | | KonIQ-10k | | Average | |
|---|---|---|---|---|---|---|---|
| | | SRCC | PLCC | SRCC | PLCC | SRCC | PLCC |
| DBCNN Zhang et al. (2018) | SPAQ | 0.702 | 0.748 | 0.684 | 0.702 | 0.693 | 0.725 |
| MUSIQ Ke et al. (2021) | SPAQ | **0.813** | **0.798** | 0.753 | 0.680 | **0.783** | 0.739 |
| CLIPIQA+ Wang et al. (2023) | SPAQ | 0.719 | 0.755 | **0.753** | **0.777** | 0.736 | **0.766** |
| SynDR-IQA | KADID-10k | 0.713 | 0.714 | 0.727 | 0.735 | 0.720 | 0.725 |

## A.5 LIMITATIONS

While SynDR-IQA demonstrates significant improvements in synthetic-to-algorithmic generalization scenarios, there are still notable performance gaps with practical availability.We think the primary challenge stems from the fundamental difference between existing synthetic distortion patterns and algorithmic distortion characteristics. Current synthetic distortion datasets primarily focus on traditional degradation types (e.g., blur, noise, compression), which differ significantly from the complex patterns introduced by modern image processing algorithms, especially those involving deep learning-based methods.

This limitation highlights the need for synthetic distortion generation methods that can produce synthetic distortions with algorithmic distortion and other complex characteristics while maintaining controllable image quality degradation. We believe addressing this limitation through future research will be crucial for further improving the generalization capability of BIQA models across different distortion scenarios.

