# OpenReview forum: "Towards Syn-to-Real IQA: A Novel Perspective on Reshaping Synthetic Data Distributions"
_ICLR.cc/2025/Conference — Submitted to ICLR 2025_

### Official Review · Reviewer_Gos6 · 2024-10-20

**Soundness:** 3
**Presentation:** 3
**Contribution:** 3
**Rating:** 6
**Confidence:** 5

**Summary:**

This paper propose a framework to enhance syn-to-real generalization ability of BIQA models. Based on theoretical analysis, a DDCup strategy along with a DRCDown strategy are proposed to reshape the distribution of synthetic distortions, leading to improved syn-to-real generalization of a BIQA model from a data perspective. Tthe effectiveness of SynDR-IQA is empirically validated across three cross-dataset settings (synthetic-to-authentic, syntheticto-algorithmic, and synthetic-to-synthetic).

**Strengths:**

+ This work presents an novel data augmentation scheme for syn-to-real BIQA that automatically identify the most sample-worthy images from a large candidate pool.
+ Theoretical analysis is provided as a foundation of the implemented strategies.
+ Cross-dataset evaluation are conducted with three settings, corresponding to different application scenarios.

**Weaknesses:**

- The motivation of syn-to-real generalization should be better elaborated, given that existing BIQA models can already handle both syntheic and authentic distortions well with a single set of model parameters, e.g., UNIQUE, Q-align, LIQE, etc.
- In addition to ResNet-50, it's better to experiment with more advanced backbone (e.g., ViT, Swin Transformer, etc.) networks to verify the generalizability of the proposed data scheme.
- Why training the feature extractor used in DDCup for only three epochs?
- During training, one patch of resolution 224×224 from each image is sampled. How about inference ?

**Questions:**

1 Some results are inconsistent with prior work, e.g., the cross-dataset results of DBCNN on LIVEC and BID are significantly lower than that reported in [1]. The authors should double check the experiments.

[1] Uncertainty-aware blind image quality assessment in the laboratory and wild.

2 The proposed method seems also be useful in continual learning settings (e.g., Continual learning for BIQA, LIQA, Remember-and-Reuse, TSN-IQA, etc.). Some discussions are expected.

---

> ### Author Response · Authors · 2024-11-22
> **Response to Reviewer Gos6 (Part1/3)**
>
> **Weakness 1-1:** The motivation of syn-to-real generalization should be better elaborated, given that existing BIQA models can already handle both syntheic and authentic distortions well with a single set of model parameters, e.g., UNIQUE, Q-align, LIQE, etc.
>
> **Response 1-1:**
> We appreciate these concerns about the motivation.
> While existing BIQA models like UNIQUE, Q-align, and LIQE have demonstrated promising performance in handling both synthetic and authentic distortions with a single set of parameters, their practical application faces significant challenges. The performance improvement of these models depends heavily on the large amount of annotation data. When encountering new, unseen scenarios, they struggle to maintain consistent performance without substantial additional labeled data for fine-tuning.
>
> This limitation becomes particularly critical given that obtaining large-scale human annotations for real-world distortions is prohibitively expensive and time-consuming. It is impractical to create comprehensive labeled datasets for the vast variety of real-world scenarios we might encounter. **This creates a fundamental gap between laboratory performance and real-world applicability.**
>
> Therefore, our focus on improving synthetic-to-real generalization is driven by this practical constraint, aiming to develop models that can effectively transfer knowledge from easily obtainable synthetic data to real-world scenarios, thus providing a more scalable and practical solution for real-world applications.
>
> **Our approach has demonstrated significant effectiveness in this direction.** We compare our model (trained on KADID-10k) with state-of-the-art models trained on SPAQ (an authentic distortion dataset with 11k images) when testing on LIVEC and KonIQ-10k. While there is still a gap compared to models trained on authentic distortions, our method achieves competitive performance despite using only synthetic data for training.
>
> |                 |                  | LIVEC       | KonIQ-10k   | Average     |
> | --------------- | ---------------- | ----------- | ----------- | ----------- |
> | Methods         | Training Dataset | SRCC PLCC   | SRCC PLCC   | SRCC PLCC   |
> | DBCNN           | SPAQ             | 0.702 0.748 | 0.684 0.702 | 0.693 0.725 |
> | MUSIQ           | SPAQ             | 0.813 0.798 | 0.753 0.680 | 0.783 0.739 |
> | CLIPIQA+        | SPAQ             | 0.719 0.755 | 0.753 0.777 | 0.736 0.766 |
> | SynDR-IQA(ours) | KADID-10k        | 0.713 0.714 | 0.727 0.735 | 0.720 0.725 |
>
> Moreover, our work has **broader implications for real-world IQA research**:
> 1. Theoretical Insights: Our theoretical analysis of how data diversity and redundancy affect model generalization provides valuable guidance for future dataset construction and method development in both synthetic and real-world scenarios.
> 2. Joint Learning: Previous studies have shown that incorporating synthetic distortion data can enhance performance in joint learning settings with real-world data [1]. Our improved synthetic data distribution could potentially further boost performance in such scenarios.
> 3. Incremental Learning: Our proposed DDCUp and DRCDown strategies offer valuable insights into incremental learning methods, particularly on how to maintain distribution consistency and optimize data sampling.
>
> These aspects demonstrate that our work, while focused on synthetic-to-real transfer, has significant implications for advancing real-world IQA research more broadly.
>
> [1] Zhang W, Ma K, Zhai G, et al. Uncertainty-aware blind image quality assessment in the laboratory and wild[J]. IEEE Transactions on Image Processing, 2021, 30: 3474-3486.
>
> ---
>
> **Weakness 1-2:**
> In addition to ResNet-50, it's better to experiment with more advanced backbone (e.g., ViT, Swin Transformer, etc.) networks to verify the generalizability of the proposed data scheme.
>
> **Response 1-2:**
> We appreciate this insightful comment. **We have conducted additional experiments using Swin-Transformer as the backbone.** The experimental results show that the proposed framework is also effective on Swin-Transformer, which verifies the generality of the proposed method. **All results has been added in Appendix 3.**
>
> |           | LIVEC           | KonIQ-10k       | BID             | Average         |
> | --------- | --------------- | --------------- | --------------- | --------------- |
> | Methods   | SRCC PLCC       | SRCC PLCC       | SRCC PLCC       | SRCC PLCC       |
> | Baseline  | 0.620 0.641     | 0.600 0.625     | 0.729 0.684     | 0.649 0.650     |
> | SynDR-IQA | **0.670 0.693** | **0.719 0.752** | **0.777 0.750** | **0.721 0.731** |

---

> ### Author Response · Authors · 2024-11-22
> **Response to Reviewer Gos6 (Part2/3)**
>
> **Weakness 2:** Why training the feature extractor used in DDCup for only three epochs?
>
> **Response 2:**
> Thanks to the reviewer for the valuable comment.
> DDCUp aims to enrich the training data with diverse visual content while preserving the underlying content distribution of the original training set. For this purpose, we initially wanted the feature extractor to focus primarily on content features rather than distortion patterns. Extended distortion-aware feature learning might bias the model towards certain distortion patterns, potentially affecting content selection for images with similar distortion characteristics. Through further experiments, we discovered that the feature extractor pre-trained on ImageNet without fine-tuning produces the same content selection compared to the fine-tuned version.
> Based on these findings, **we have updated our implementation to directly use the ImageNet pre-trained model as the feature extractor**, which simplifies the pipeline while maintaining effectiveness.
>
>
> ---
> **Weakness 3:** During training, one patch of resolution 224×224 from each image is sampled. How about inference?
>
> **Response 3:** During inference, we sample five patches per image and use their average prediction as the final output, as described in Section 4.1. This setup follows the same protocol as DGQA [1] and is a common practice, as also adopted by other methods like HyperIQA [2], KGANet[3].
>
> ---
> **Question 1:** Some results are inconsistent with prior work, e.g., the cross-dataset results of DBCNN on LIVEC and BID are signiìcantly lower than that reported in [1]. The authors should double check the experiments.
> [1] Uncertainty-aware blind image quality assessment in the laboratory and wild.
>
> **Response 1:**
> We sincerely appreciate the reviewer's careful examination of our results. The DBCNN results in our original manuscript were cited from [2]. To ensure result reliability, we have re-implemented DBCNN's cross-dataset evaluation using their official PyTorch implementation ([https://github.com/zwx8981/DBCNN-PyTorch/tree/master](https://github.com/zwx8981/DBCNN-PyTorch/tree/master)). **The results have been updated in the Table I.** Below we present a comprehensive comparison between the results reported in [1], our reproduction of DBCNN and our proposed method.
>
> |                    | LIVEC           | KonIQ-10k           | BID                 | Average             |
> | ------------------ | --------------- | ------------------- | ------------------- | ------------------- |
> | Methods            | SRCC PLCC       | SRCC PLCC           | SRCC PLCC           | SRCC PLCC           |
> | DBCNN (from [1])   | 0.531 -         | -        -          | 0.602 -             | -         -         |
> | DBCNN (Reproduced) | 0.572 0.589     | 0.639 0.618         | 0.620 0.609         | 0.613 0.606         |
> | SynDR-IQA(ours)    | **0.713** 0.714 | **0.727** **0.735** | **0.788** **0.764** | **0.743** **0.737** |
>
> Our reproduced results show slightly different but comparable performance to those reported in [1]. More importantly, the significant performance gap between our method and DBCNN remains consistent, which further validates the effectiveness of our approach.
>
> [1] Zhang W, Ma K, Zhai G, et al. Uncertainty-aware blind image quality assessment in the laboratory and wild[J]. IEEE Transactions on Image Processing, 2021, 30: 3474-3486.
> [2] Li X, Lu Y, Chen Z. Freqalign: Excavating perception-oriented transferability for blind image quality assessment from a frequency perspective[J]. IEEE Transactions on Multimedia, 2023.

---

> ### Author Response · Authors · 2024-11-22
> **Response to Reviewer Gos6 (Part3/3)**
>
> **Question 2:** The proposed method seems also be useful in continual learning settings (e.g., Continual learning for BIQA, LIQA, Remember-andReuse, TSN-IQA, etc.). Some discussions are expected.
>
> **Response 2:**
> We appreciate this insightful observation. Indeed, our framework shows promising potential for continual learning scenarios in IQA tasks, primarily in three aspects:
> 1. Since different tasks/datasets often exhibit distribution discrepancies, the theoretical framework we developed for analyzing how sample diversity and redundancy may help for guiding the selection of samples during the incremental learning process.
> 2. Our methodology offers valuable insights for continual learning approaches. DDCUp strategy demonstrates how to introduce diverse data while maintaining distribution consistency, potentially addressing catastrophic forgetting. DRCDown strategy could improve data sampling during the incremental process by managing redundant samples and balancing sample density, particularly useful when incorporating new tasks or domains.
> 3. Our method could benefit continual learning scenarios where synthetic data is involved. Previous studies have shown that incorporating synthetic distortion data can enhance performance in joint learning settings with real-world data [1]. Our enhanced synthetic data distribution could further improve this synergy while reducing potential interference with other scenario data.
>
> These aspects demonstrate that while our work focuses on synthetic-to-real transfer, its underlying principles and strategies could be valuable for advancing continual learning in IQA tasks. Future work could explore specific adaptations of our framework for various continual learning protocols and scenarios.
>
> [1] Zhang W, Ma K, Zhai G, et al. Uncertainty-aware blind image quality assessment in the laboratory and wild[J]. IEEE Transactions on Image Processing, 2021, 30: 3474-3486.

---

> ### Author Response · Authors · 2024-11-27
> **Looking forward to your feedback**
>
> Dear Reviewer Gos6,
>
> Thank you for your valuable and constructive feedback, which has greatly helped improve our work.
>
> With the paper revision deadline approaching, we would be grateful if you could review our response and let us know if our explanations and revisions have adequately addressed your concerns. Any further comments or suggestions you may have would be greatly valued.
>
> We sincerely appreciate your time and expertise and look forward to your feedback.
>
> Best regards,
> The Authors

---

> ### Comment · Reviewer_Gos6 · 2024-11-28
>
> Thanks for the responses, which partially address my concerns. I will keep the rating.

---

### Official Review · Reviewer_kTZx · 2024-11-03

**Soundness:** 2
**Presentation:** 3
**Contribution:** 2
**Rating:** 6
**Confidence:** 4

**Summary:**

This paper analyzes the response of ResNet to images of varying quality, noting that the model emphasizes content in high-quality images while focusing more on degradation in lower-quality images. From these findings, the paper identifies diversity and redundancy in samples as two crucial factors affecting generalization ability. Correspondingly, it proposes an optimized strategy for constructing the training dataset, achieving promising results in cross-dataset evaluations.

**Strengths:**

1. Enhancing generalization is a critical issue in IQA, and this paper addresses it by proposing two effective strategies based on theoretical analysis, achieving improvement over baseline in quantitative metrics.
2. The Fig1 visualization is clear, and the paper provides ablation studies. The writing is also accessible and easy to understand.

**Weaknesses:**

1. In real-world IQA, the primary challenge in generalization is often the variety in degradation types, rather than content diversity alone. As illustrated in Figure 1, for heavily degraded images, the model tends to focus more on the degradation type. The proposed method adds more high-quality reference images but does not fundamentally address this issue.
2. Although Figure 1 reveals some aspects of ResNet's performance, many IQA methods utilize other models, such as CLIP. Thus, the conclusions drawn in Figure 1 may not be universally applicable. A more comprehensive analysis across various models would enhance the validity of the findings.
3. The use of a new dataset as a reference may lead to comparisons that are not entirely fair with other methods.
4. The paper does not compare its method against some of the most recent SOTA approaches in the field, such as CLIPIQA and Q-Align. While the proposed method shows some improvements, fundamental generalization is often best achieved by exposing the model to more diverse training data, covering different degradation types, content variations, etc. Relying solely on simulated degradation effects has its limitations.
5. The paper evaluates methods such as SR and denoising using SRCC and PLCC metrics. However, common metrics in these fields include PSNR, SSIM, and LPIPS, which are not compared here. Considering there may be no available reference images, non-reference metrics like MANIQA, MUSIQ, and CLIPIQA are also widely used.
6. Only quantitative results are provided, with no visual examples. As IQA metrics often diverge significantly from human perception, presenting visualizations with corresponding scores would help illustrate the effectiveness of the proposed approach.

**Questions:**

1. Why do Algorithms 1 and 2 use the median as a threshold? Is there any theoretical or empirical basis for this choice?
2. Does the proposed method improve performance on images with different degradation types, such as low light, noise, defocus, or more complex degradations? Please provide relevant results and analyses.

---

> ### Author Response · Authors · 2024-11-22
> **Response to Reviewer kTZx (Part1/4)**
>
> **Weakness 1:** In real-world IQA, the primary challenge in generalization is often the variety in degradation types, rather than content diversity alone. As illustrated in Figure 1, for heavily degraded images, the model tends to focus more on the degradation type. The proposed method adds more high-quality reference images but does not fundamentally address this issue.
>
> **Response 1:**
> Thanks to the reviewer for the insightful observation. We agree that generating better synthetic degradation types will be an important direction for future work to further improve syn-to-real generalization.
>
> However, based on our analysis of Figure 1, we respectfully suggest that content diversity, rather than degradation variety, represents the primary challenge. Our analysis shows that while low-quality images do cluster by degradation types, they are tightly concentrated in the central region with relatively small inter-cluster distances, indicating the model can reliably identify poor quality regardless of specific degradation types. In contrast, high-quality images form highly discrete clusters in the peripheral region, with inter-cluster distances even larger than the separation between high and low-quality regions, suggesting significant uncertainty in handling different image contents. This observation aligns well with empirical experience in IQA tasks.
>
> To address this challenge, our method incorporates several complementary strategies. The core component is our DDCUp strategy, which maintains distribution consistency while enriching content variety.
>
> Meanwhile, **we haven't overlooked the importance of degradation handling** - our approach adopts DGQA's degradation type selection for more appropriate distortion types, alongside the DRCDown strategy that reduces redundant clusters (both distortion- and content- based) and helps learn more balanced representations. And in Figure 3, it can be found that our model gets more continuous quality representations. Besides, the comprehensive experiments across various datasets also validate the effectiveness of our approach.
>
> ---
> **Weakness 2:** Although Figure 1 reveals some aspects of ResNet's performance, many IQA methods utilize other models, such as CLIP. Thus, the conclusions drawn in Figure 1 may not be universally applicable. A more comprehensive analysis across various models would enhance the validity of the findings.
>
> **Response 2:**
> We appreciate this insightful comment. We would like to clarify that the observed phenomenon stems from the inherent properties of synthetic datasets (insufficient diversity and excessive redundancy), independent of model architectures. **We have conducted additional experiments using Swin-Transformer as the backbone**, which shows **similar feature distribution patterns** to ResNet. More importantly, our method shows even more significant improvements for Swin-Transformer. **All results has been added in Appendix 3.**
>
> |           | LIVEC           | KonIQ-10k       | BID             | Average         |
> | --------- | --------------- | --------------- | --------------- | --------------- |
> | Methods   | SRCC PLCC       | SRCC PLCC       | SRCC PLCC       | SRCC PLCC       |
> | Baseline  | 0.620 0.641     | 0.600 0.625     | 0.729 0.684     | 0.649 0.650     |
> | SynDR-IQA | **0.670 0.693** | **0.719 0.752** | **0.777 0.750** | **0.721 0.731** |

---

> ### Author Response · Authors · 2024-11-22
> **Response to Reviewer kTZx (Part2/4)**
>
> **Weakness 3:** The use of a new dataset as a reference may lead to comparisons that are not entirely fair with other methods.
>
> **Response 3:**
> We appreciate the reviewer's concern. While our method does introduce additional data, we want to emphasize that **these data require no manual annotation** (labels are generated through the proposed Algorithm 2). Utilizing unlabeled data is also common in the compared methods (e.g., RankIQA, DBCNN, etc.). More importantly, our ablation studies (Table 3) demonstrate that **simply adding more data actually leads to performance degradation**, which clearly demonstrates the contribution of our proposed approach to the performance improvement.
>
> ---
> **Weakness 4-1:** The paper does not compare its method against some of the most recent SOTA approaches in the field, such as CLIPIQA and QAlign.
>
> **Response 4-1:**
> We appreciate the reviewer's suggestion regarding comparisons with SOTA methods. **We have expanded our experimental comparisons to include CLIPIQA, CLIPIQA+, and Qalign in Table 1.** The results for CLIPIQA and CLIPIQA+ are obtained by running the authors' source code, while the results of Qalign are from their original paper. The comparative results are shown below:
>
> |                 | LIVEC           | KonIQ-10k           | BID                 | Average             |
> | --------------- | --------------- | ------------------- | ------------------- | ------------------- |
> | Methods         | SRCC PLCC       | SRCC PLCC           | SRCC PLCC           | SRCC PLCC           |
> | CLIPIQA         | 0.643 0.629     | 0.684 0.702         | 0.627 0.581         | 0.651 0.637         |
> | CLIPIQA+        | 0.512 0.543     | 0.511 0.515         | 0.474 0.442         | 0.499 0.500         |
> | Q-Align         | 0.702 **0.744** | 0.668 0.665         | -        -          | -        -          |
> | SynDR-IQA(ours) | **0.713** 0.714 | **0.727** **0.735** | **0.788** **0.764** | **0.743** **0.737** |
>
> As shown in the table, our method consistently outperforms these SOTA approaches across different datasets, demonstrating its superior performance.

---

> ### Author Response · Authors · 2024-11-22
> **Response to Reviewer kTZx (Part3/4)**
>
> **Weakness 4-2:** While the proposed method shows some improvements, fundamental generalization is often best achieved by exposing the model to more diverse training data, covering different degradation types, content variations, etc. Relying solely on simulated degradation effects has its limitations.
>
> **Response 4-2:**
> Thanks to the reviewer for the comment. We agree with the statement that "fundamental generalization is often best achieved by exposing the model to more diverse training data." **Indeed, our method makes significant contributions in this direction.** We designed a novel pseudo-label generation scheme (Algorithm 2) to introduce additional data cost-effectively. Beyond merely incorporating more data, we theoretically analyzed how data diversity and redundancy affect model generalization, providing guidance on what kind of data is needed (Ablation experiments have demonstrated that simply introducing data does not bring the expected performance improvement). Based on this analysis, we proposed DDCUp to address the challenge of introducing beneficial data and DRCDown to filter out potentially harmful samples.
>
> Regarding the reviewer's **concern about synthetic distortion diversity**, DGQA has demonstrated that simply incorporating more distortion types may not necessarily lead to better performance and could even have negative effects. While we acknowledge that better distortion synthesis methods could help create more suitable degradations and further improve model performance, we look forward to future research advancing this direction.
>
> Regarding the reviewer's **concern about the limitations of synthetic distortions**, our approach has demonstrated significant effectiveness in this direction. We compare our model (trained on KADID-10k) with state-of-the-art models trained on SPAQ (an authentic distortion dataset with 11k images) when testing on LIVEC and KonIQ-10k. While there is still a gap compared to models trained on authentic distortions, our method achieves competitive performance despite using only synthetic data for training.
>
> |                 |                  | LIVEC       | KonIQ-10k   | Average     |
> | --------------- | ---------------- | ----------- | ----------- | ----------- |
> | Methods         | Training Dataset | SRCC PLCC   | SRCC PLCC   | SRCC PLCC   |
> | DBCNN           | SPAQ             | 0.702 0.748 | 0.684 0.702 | 0.693 0.725 |
> | MUSIQ           | SPAQ             | 0.813 0.798 | 0.753 0.680 | 0.783 0.739 |
> | CLIPIQA+        | SPAQ             | 0.719 0.755 | 0.753 0.777 | 0.736 0.766 |
> | SynDR-IQA(ours) | KADID-10k        | 0.713 0.714 | 0.727 0.735 | 0.720 0.725 |
>
> Moreover, our work has **broader implications for real-world IQA research**:
> 1. Theoretical Insights: Our theoretical analysis of how data diversity and redundancy affect model generalization provides valuable guidance for future dataset construction and method development in both synthetic and real-world scenarios.
> 2. Joint Learning: Previous studies have shown that incorporating synthetic distortion data can enhance performance in joint learning settings with real-world data [1]. Our improved synthetic data distribution could potentially further boost performance in such scenarios.
> 3. Incremental Learning: Our proposed DDCUp and DRCDown strategies offer valuable insights into incremental learning methods, particularly on how to maintain distribution consistency and optimize data sampling.
>
> These aspects demonstrate that our work, while focused on synthetic-to-real transfer, has significant implications for advancing real-world IQA research more broadly.
>
> [1] Zhang W, Ma K, Zhai G, et al. Uncertainty-aware blind image quality assessment in the laboratory and wild[J]. IEEE Transactions on Image Processing, 2021, 30: 3474-3486.
>
> ---
>
> **Weakness 5:** The paper evaluates methods such as SR and denoising using SRCC and PLCC metrics. However, common metrics in these ìelds include PSNR, SSIM, and LPIPS, which are not compared here. Considering there may be no available reference images, nonreference metrics like MANIQA, MUSIQ, and CLIPIQA are also widely used.
>
> **Response 5:**
> We appreciate the reviewer's suggestion, but we would like to clarify that there seems to be a misunderstanding. Our work focuses on Image Quality Assessment (IQA) rather than developing SR or denoising methods.
>
> The experiments in Table 2 are conducted on the algorithm-distorted subset of PIPAL (**an IQA dataset**). The goal is to evaluate our model's capability to predict the perceptual quality of algorithm-processed images (e.g., those processed by SR, denoising, etc.) after being trained solely on the synthetic distortion dataset.
>
> Therefore, SRCC and PLCC are the appropriate metrics as they measure the correlation between predicted quality scores and human judgments, which is the standard evaluation protocol in IQA research.

---

> ### Author Response · Authors · 2024-11-22
> **Response to Reviewer kTZx (Part4/4)**
>
> **Weakness 6:** Only quantitative results are provided, with no visual examples. As IQA metrics often diverge significantly from human perception, presenting visualizations with corresponding scores would help illustrate the effectiveness of the proposed approach.
>
> **Response 6:**
> We appreciate the reviewer's suggestion. However, for cross-dataset IQA works [1,2], direct visual comparison between predictions and ground truth is typically not applicable due to **inconsistent quality scales across different datasets** [1].
>
> For IQA metrics, researchers typically **focus more on how well they reflect the relative quality relationships between images** rather than the specific predicted values. This is why SRCC, rather than RMSE, is commonly used as the primary metric for evaluating IQA models.
>
> Nevertheless, **we have included visual examples comparing predictions with ground truth in our Appendix 2 for reference**.
>
> [1] Zhang W, Ma K, Zhai G, et al. Uncertainty-aware blind image quality assessment in the laboratory and wild[J]. IEEE Transactions on Image Processing, 2021, 30: 3474-3486.
> [2] Wang J, Chan K C K, Loy C C. Exploring clip for assessing the look and feel of images[C]//Proceedings of the AAAI Conference on Artificial Intelligence. 2023, 37(2): 2555-2563.
>
> ---
> **Question 1:** Why do Algorithms 1 and 2 use the median as a threshold? Is there any theoretical or empirical basis for this choice?
>
> **Response 1:**
> We appreciate this question about the choice of median as a threshold.
> The choice serves dual purposes in our framework: it prevents the introduction of highly similar images that could increase redundancy while avoiding overly dissimilar images that might introduce noise into the learning process. Unlike fixed thresholds, the median provides an adaptive mechanism, eliminating the need for manual threshold tuning and naturally adapting to different datasets.
> Besides, the choice is further supported by empirical evidence, as median-based thresholding has been successfully employed in various resampling frameworks [1,2].
>
> [1] Torgo L, Ribeiro R P, Pfahringer B, et al. Smote for regression[C]//Portuguese conference on artificial intelligence. Berlin, Heidelberg: Springer Berlin Heidelberg, 2013: 378-389.
> [2] Branco P, Torgo L, Ribeiro R P. SMOGN: a pre-processing approach for imbalanced regression[C]//First international workshop on learning with imbalanced domains: Theory and applications. PMLR, 2017: 36-50.
>
> ---
> **Question 2:** Does the proposed method improve performance on images with different degradation types, such as low light, noise, defocus, or more complex degradations? Please provide relevant results and analyses.
>
> **Response 2:**
> We appreciate this question about performance on different degradation types.
> **The mentioned degradation types (low light, noise, defocus) and their complex combinations are prevalent in authentic distortion datasets (although these degradations are not explicitly labeled).** Our method shows significant improvements on these datasets (in Table I). Additionally, our method also shows a significant improvement on complex algorithmic distortions (in Table II).
> **Besides, in the Appendix 2, we have added visual examples including the images with low light, noise, defocus and other complex degradations.** These examples provide qualitative evidence of our method's capability to handle diverse degradation types.
>
> Our framework has already demonstrated comprehensive validation across a broad range of scenarios, including three cross-domain settings (synthetic-to-authentic, synthetic-to-algorithmic, and synthetic-to-synthetic) and seven datasets. The consistent performance improvements across this wide range of scenarios and degradation types strongly support our method's generalization capability and robustness.
>
> We hope these results can comprehensively address the concern about our method's effectiveness across different degradation types.

---

> > ### Comment · Reviewer_kTZx · 2024-11-27
> > **Response to Authors**
> >
> > The results in Table 4-2 raise a question: why is the performance of the proposed methods significantly behind MUSIQ? Additionally, could more fair comparisons be provided by using the same training data? Although Figure 4 in the appendix shows that the proposed method aligns well with human perception, it would be beneficial to include the results of other methods and quantitative results on a large test dataset.

---

> > > ### Author Response · Authors · 2024-11-27
> > >
> > > **Question 1:** The results in Table 4-2 raise a question: why is the performance of the proposed methods significantly behind MUSIQ? Additionally, could more fair comparisons be provided by using the same training data?
> > >
> > > **Response 1:**
> > > Thank you for these important questions. We would like to clarify that the performance difference stems from two fundamental aspects about the different training datasets:
> > > 1. **Distribution Gap Difference**: SPAQ and LIVEC are both authentic distortion datasets  with similar distributions. In contrast, KADID-10k is a synthetic distortion dataset with a significant distribution gap from LIVEC.
> > > 2. **Content Diversity Difference**: SPAQ contains rich content diversity (11125 images), enabling networks to learn continuous quality representations that generalize well. KADID-10k has limited content diversity (only 81 reference images), which leads to more discrete representations with poorer generalization performance.
> > >
> > > We compare the generalization performance of our method with MUSIQ trained separately on SPAQ and KADID-10k (**the performances of MUSIQ trained on KADID-10k are shown in Table 1 of our paper**). The results demonstrate that, **when both trained on KADID-10k, SynDR-IQA significantly outperforms MUSIQ**. Although a gap remains compared to real-to-real transfer, our method substantially reduces this gap, marking a major advancement in synthetic-to-real transfer for IQA.
> > >
> > > |                 |                  | LIVEC       | KonIQ-10k   | Average     |
> > > | --------------- | ---------------- | ----------- | ----------- | ----------- |
> > > | Methods         | Training Dataset | SRCC PLCC   | SRCC PLCC   | SRCC PLCC   |
> > > | MUSIQ           | SPAQ             | 0.813 0.798 | 0.753 0.680 | 0.783 0.739 |
> > > | MUSIQ           | KADID-10k        | 0.517 0.524 | 0.554 0.573 | 0.536 0.549 |
> > > | SynDR-IQA(ours) | KADID-10k        | 0.713 0.714 | 0.727 0.735 | 0.720 0.725 |
> > >
> > > ---
> > >
> > > **Question 2:** Although Figure 4 in the appendix shows that the proposed method aligns well with human perception, it would be beneficial to include the results of other methods and quantitative results on a large test dataset.
> > >
> > > **Response 2:**
> > > Thank you for the valuable suggestions. **We have updated Figure 4 in the Appendix to include a comparison of predicted results between our proposed method and the SOTA method CLIP-IQA.** The results demonstrate that our predictions show significantly better alignment with human perception.
> > >
> > > Regarding the suggestion for **quantitative results on a large test dataset**, we would like to clarify that **our paper already includes comprehensive quantitative analyses across 8 commonly used datasets in 3 different scenarios** (as detailed in Tables 1, 2, and 3 in the manuscript). The quantitative results for LIVEC, the complete dataset that contains all the visual examples, can also be found in Table 1. These results collectively verify the effectiveness of our proposed method.
> > >
> > > ---
> > >
> > > We sincerely thank you for your insightful feedback and the time you have dedicated to reviewing our work. We hope that our responses have adequately addressed your concerns and provided the necessary clarification. We remain open to further discussion and are happy to offer additional explanations if needed.

---

> > > > ### Comment · Reviewer_kTZx · 2024-12-01
> > > > **Response to Authors**
> > > >
> > > > Thanks for the response. Most of my concerns are addressed. I would like to raise my rating to borderline accept.

---

### Official Review · Reviewer_zhhN · 2024-11-03

**Soundness:** 2
**Presentation:** 3
**Contribution:** 3
**Rating:** 6
**Confidence:** 3

**Summary:**

This paper proposes a SynDR-IQA framework, which aims to improve the generalization ability of BIQA models by adjusting the distribution of synthetic data, i.e., distribution-aware diverse content up-sampling and density-aware redundant cluster down-sampling. Experiments across three cross-dataset settings, including synthetic-to-authentic, synthetic-to-algorithmic, and synthetic-to-synthetic, demonstrate the effectiveness of SynDR-IQA.

**Strengths:**

1. Improving the generalization of deep models by adjusting the distribution of synthetic data is a promising research direction.

2. A theoretical derivation of the effect of sample diversity and redundancy on the generalization error was made in the paper, thus laying a theoretical foundation for the proposal of SynDR-IQA.

3. Multiple experiments across dataset settings validate the effectiveness of the method. Furthermore, as a data-driven approach, SynDR-IQA can be used in conjunction with existing model-based approaches without increasing inference costs.

**Weaknesses:**

1. The proposed method may be more complex in practice, requiring additional steps to adjust the data distribution, which would increase the difficulty of implementation. In other words, although SynDR-IQA does not increase the inference cost, it requires more computational resources in the training phase, especially for large-scale datasets.

2. Although this paper proposes a BIQA method based on data distribution adjustment, more explanations and analyses need to be provided on why it improves generalization.

3. Further tests to verify the robustness of SynDR-IQA when addressing different types and degrees of image distortion should be supplemented with experiments.

**Questions:**

Please refer to the weakness.

---

> ### Author Response · Authors · 2024-11-22
> **Response to Reviewer zhhN**
>
> **Weakness 1:** The proposed method may be more complex in practice, requiring additional steps to adjust the data distribution, which would increase the difficulty of implementation. In other words, although SynDR-IQA does not increase the inference cost, it requires more computational resources in the training phase, especially for large-scale datasets.
>
> **Response 1:**
> We acknowledge the reviewer's concern about computational complexity and would like to address it from several perspectives:
> 	1. Current IQA datasets have **limited training samples**, which naturally constrains the computational demands. Besides, our DRCDown strategy actually **helps reduce data redundancy in the original dataset**.
> 	2. While our method requires some additional computational resources during training, we believe this is **a worthwhile trade-off** given the significant performance improvements achieved during inference with the same computational cost. Notably, compared to methods requiring large-scale pre-training, our approach introduces only **limited computational overhead**.
> 	3. To support practical implementation, we will make our complete source code and all associated resources publicly available.
>
> ---
> **Weakness 2:** Although this paper proposes a BIQA method based on data distribution adjustment, more explanations and analyses need to be provided on why it improves generalization.
>
> **Response 2:**
> We appreciate the reviewer's concern. **In Section 3.2, we provide theoretical analysis through Theorem 1**, which establishes the generalization error bound for datasets with redundant samples. This bound is determined by three key factors: the number of *i.i.d.* samples $m$, redundancy heterogeneity $\eta$, and total sample size $n$.
>
> Our analysis reveals that increasing content diversity $m$ and reducing redundancy heterogeneity $\eta$ can effectively lower the generalization error bound, while simply adding redundant samples $n$ may increase $\eta$ and potentially worsen generalization.
>
> This theoretical foundation directly guides the design of our framework: DDCUp strategy increases content diversity (increasing $m$) while DRCDown strategy balances sample density (reducing $\eta$), jointly improving model generalization. The effectiveness of these theoretically-motivated strategies is validated by our extensive experimental results across different cross-dataset settings.
>
> ---
> **Weakness 3:** Further tests to verify the robustness of SynDR-IQA when addressing different types and degrees of image distortion should be supplemented with experiments.
>
> **Response 3:**
> We appreciate the reviewer's suggestion. However, we would like to clarify that our primary research focus is not on improving performance across different synthetic distortion types and intensities (authentic distortion data usually does not have clear distortion types and degrees), which can be readily addressed by incorporating more corresponding synthetic distortion training data (like RankIQA). Instead, our work tackles a more fundamental challenge: enhancing the generalization capability from synthetic to real-world distortions by reshaping data distributions.
>
> Nevertheless, **our framework has already demonstrated comprehensive validation across a broad range of scenarios,** including three cross-domain settings (synthetic-to-authentic, synthetic-to-algorithmic, and synthetic-to-synthetic) and seven datasets. These results collectively validate that our approach effectively improves model generalization.

---

> ### Author Response · Authors · 2024-11-27
> **Looking forward to your feedback**
>
> Dear Reviewer zhhN,
>
> Thank you for your valuable and constructive feedback, which has greatly helped improve our work.
>
> With the paper revision deadline approaching, we would be grateful if you could review our response and let us know if our explanations and revisions have adequately addressed your concerns. Any further comments or suggestions you may have would be greatly valued.
>
> We sincerely appreciate your time and expertise and look forward to your feedback.
>
> Best regards,
> The Authors

---

> > ### Comment · Reviewer_zhhN · 2024-11-29
> >
> > Thank you for your reply! It addresses my concerns to some extent. The explanation would become more adequate and robust if it had been accompanied by some experimental illustrations. Based on the responses so far, I will maintain my original rating.

---

### Official Review · Reviewer_aBSv · 2024-11-04

**Soundness:** 3
**Presentation:** 2
**Contribution:** 3
**Rating:** 6
**Confidence:** 3

**Summary:**

1. **Feature Distribution Issue**: Models trained on synthetic data show discrete and clustered features, limiting generalization due to synthetic data distribution.

2. **SynDR-IQA Framework**: Introduces DDCUp and DRCDown strategies to enhance diversity and balance density, reshaping data for better generalization.

3. **Validation**: SynDR-IQA proves effective across cross-dataset scenarios and integrates with existing models without extra inference costs.

**Strengths:**

1. **Innovative Approach**: Proposes SynDR-IQA, a novel framework that reshapes synthetic data distribution to improve BIQA generalization.

2. **Effective Strategies**: Introduces DDCUp and DRCDown, two strategies that enhance data diversity and reduce redundancy, addressing core issues in synthetic datasets.

3. **Strong Experimental Validation**: Demonstrates effectiveness across multiple cross-dataset settings, showing that SynDR-IQA enhances performance without increasing inference costs.

**Weaknesses:**

1. **lack of IQA result images**. Visual examples comparing predictions with ground truth would clarify SynDR-IQA's impact on quality assessment.
2. **Synthetic Data Focus**: Primarily targets synthetic data limitations, which may limit applicability to real-world data.
3. **Limited Analysis on Failure Cases**: The paper lacks discussion on SynDR-IQA's potential limitations or specific cases where it may underperform, such as certain distortion types or datasets. A balanced evaluation could clarify the framework's boundaries.

**Questions:**

1. **Lack of Comparisons with Advanced Methods**: : The paper does not include comparisons with state-of-the-art methods like Q-align[1], LIQE[2], CLIPIQA[3], and TOPIQ[4], which would provide a clearer benchmark of SynDR-IQA’s performance.

2. **Complex Implementation**: The DDCUp and DRCDown strategies, while theoretically valuable, add complexity in implementation. Practical adoption may be challenging without detailed instructions or code, possibly limiting its use in broader BIQA applications.

﻿3. **Synthetic Data Focus**: The framework mainly addresses synthetic-to-real transitions, which may limit its applicability and effectiveness on purely real-world distortions where data characteristics and distortion patterns differ substantially from synthetic ones.




[1] Wu, Haoning, et al. "Q-align: Teaching lmms for visual scoring via discrete text-defined levels." arXiv preprint arXiv:2312.17090 (2023).

[2] Zhang, Weixia, et al. "Blind image quality assessment via vision-language correspondence: A multitask learning perspective." Proceedings of the IEEE/CVF conference on computer vision and pattern recognition. 2023.

[3] Wang, Jianyi, Kelvin CK Chan, and Chen Change Loy. "Exploring clip for assessing the look and feel of images." Proceedings of the AAAI Conference on Artificial Intelligence. Vol. 37. No. 2. 2023.

[4] Chen, Chaofeng, et al. "Topiq: A top-down approach from semantics to distortions for image quality assessment." IEEE Transactions on Image Processing (2024).

---

> ### Author Response · Authors · 2024-11-22
> **Response to Reviewer aBSv (Part1/2)**
>
> **Weakness 1:** lack of IQA result images. Visual examples comparing predictions with ground truth would clarify SynDR-IQA's impact on quality assessment.
>
> **Response 1:**
> We appreciate the reviewer's suggestion. However, for cross-dataset IQA works [1,2], direct visual comparison between predictions and ground truth is typically not applicable due to **inconsistent quality scales across different datasets** [1].
>
> For IQA metrics, researchers typically **focus more on how well they reflect the relative quality relationships between images** rather than the specific predicted values. This is why SRCC, rather than RMSE, is commonly used as the primary metric for evaluating IQA models.
>
> Nevertheless, **we have included visual examples comparing predictions with ground truth in our Appendix 2 for reference**.
>
> [1] Zhang W, Ma K, Zhai G, et al. Uncertainty-aware blind image quality assessment in the laboratory and wild[J]. IEEE Transactions on Image Processing, 2021, 30: 3474-3486.
> [2] Wang J, Chan K C K, Loy C C. Exploring clip for assessing the look and feel of images[C]//Proceedings of the AAAI Conference on Artificial Intelligence. 2023, 37(2): 2555-2563.
>
> ---
> **Weakness 2 & Question 3: Synthetic Data Focus: Primarily targets synthetic data limitations, which may limit applicability to real-world data. / The framework mainly addresses synthetic-to-real transitions, which may limit its applicability and effectiveness on purely real-world distortions where data characteristics and distortion patterns differ substantially from synthetic ones.
>
> **Response 2:**
> We appreciate these concerns about synthetic data limitations. However, we want to emphasize that **our work is specifically designed to address real-world IQA challenges**. The **core motivation** stems from a practical constraint: obtaining large-scale human annotations for real-world distortions is prohibitively expensive and time-consuming, making it impractical to create comprehensive labeled datasets for various real-world scenarios. This is precisely why we focus on improving synthetic-to-real generalization as a practical solution.
>
> Our approach has demonstrated significant effectiveness in this direction. We compare our model (trained on KADID-10k) with state-of-the-art models trained on SPAQ (an authentic distortion dataset with 11k images) when testing on LIVEC and KonIQ-10k. While there is still a gap compared to models trained on authentic distortions, our method achieves competitive performance despite using only synthetic data for training.
>
> |                 |                  | LIVEC       | KonIQ-10k   | Average     |
> | --------------- | ---------------- | ----------- | ----------- | ----------- |
> | Methods         | Training Dataset | SRCC PLCC   | SRCC PLCC   | SRCC PLCC   |
> | DBCNN           | SPAQ             | 0.702 0.748 | 0.684 0.702 | 0.693 0.725 |
> | MUSIQ           | SPAQ             | 0.813 0.798 | 0.753 0.680 | 0.783 0.739 |
> | CLIPIQA+        | SPAQ             | 0.719 0.755 | 0.753 0.777 | 0.736 0.766 |
> | SynDR-IQA(ours) | KADID-10k        | 0.713 0.714 | 0.727 0.735 | 0.720 0.725 |
>
> Moreover, our work has broader implications for real-world IQA research:
> 1. Theoretical Insights: Our theoretical analysis of how data diversity and redundancy affect model generalization provides valuable guidance for future dataset construction and method development in both synthetic and real-world scenarios.
> 2. Joint Learning: Previous studies have shown that incorporating synthetic distortion data can enhance performance in joint learning settings with real-world data [1]. Our improved synthetic data distribution could potentially further boost performance in such scenarios.
> 3. Incremental Learning: Our proposed DDCUp and DRCDown strategies offer valuable insights into incremental learning methods, particularly on how to maintain distribution consistency and optimize data sampling.
>
> These aspects demonstrate that our work, while focused on synthetic-to-real transfer, has significant implications for advancing real-world IQA research more broadly.
>
> [1] Zhang W, Ma K, Zhai G, et al. Uncertainty-aware blind image quality assessment in the laboratory and wild[J]. IEEE Transactions on Image Processing, 2021, 30: 3474-3486.

---

> ### Author Response · Authors · 2024-11-22
> **Response to Reviewer aBSv (Part2/2)**
>
> **Weakness 3:** Limited Analysis on Failure Cases: The paper lacks discussion on SynDR-IQA's potential limitations or specific cases where it may underperform, such as certain distortion types or datasets. A balanced evaluation could clarify the framework's boundaries.
>
> **Response 3:**
> Thanks to the reviewer for the insightful suggestion. **We have add the discussion about limitations of our current work in Appendix 5.**
>
> While SynDR-IQA demonstrates significant improvements in synthetic-to-algorithmic generalization scenarios, there are still notable performance gaps with practical availability.
> We think the primary challenge stems from the fundamental difference between existing synthetic distortion patterns and algorithmic distortion characteristics. Current synthetic distortion datasets primarily focus on traditional degradation types (e.g., blur, noise, compression), which differ significantly from the complex patterns introduced by modern image processing algorithms, especially those involving deep learning-based methods.
>
> This limitation highlights the need for synthetic distortion generation methods that can produce synthetic distortions with algorithmic distortion and other complex characteristics while maintaining controllable image quality degradation.
>
> We believe addressing this limitation through future research will be crucial for further improving the generalization capability of BIQA models across different distortion scenarios.
>
> ---
> **Question 1:** Lack of Comparisons with Advanced Methods: : The paper does not include comparisons with state-of-the-art methods like Qalign[1], LIQE[2], CLIPIQA[3], and TOPIQ[4], which would provide a clearer benchmark of SynDR-IQA’s performance.
>
> **Response 1:**
> We appreciate the reviewer's suggestion regarding comparisons with SOTA methods. **We have expanded our experimental comparisons to include CLIPIQA, CLIPIQA+, and Qalign in Table 1.** The results for CLIPIQA and CLIPIQA+ are obtained by running the authors' source code, while the results of Qalign are from their original paper. The comparative results are shown below:
>
> |                 | LIVEC           | KonIQ-10k           | BID                 | Average             |
> | --------------- | --------------- | ------------------- | ------------------- | ------------------- |
> | Methods         | SRCC PLCC       | SRCC PLCC           | SRCC PLCC           | SRCC PLCC           |
> | CLIPIQA         | 0.643 0.629     | 0.684 0.702         | 0.627 0.581         | 0.651 0.637         |
> | CLIPIQA+        | 0.512 0.543     | 0.511 0.515         | 0.474 0.442         | 0.499 0.500         |
> | Q-Align         | 0.702 **0.744** | 0.668 0.665         | -        -          | -        -          |
> | SynDR-IQA(ours) | **0.713** 0.714 | **0.727** **0.735** | **0.788** **0.764** | **0.743** **0.737** |
>
> As shown in the table, our method consistently outperforms these SOTA approaches across different datasets, demonstrating its superior performance.
>
> ---
> **Question 2:** Complex Implementation: The DDCUp and DRCDown strategies, while theoretically valuable, add complexity in implementation. Practical adoption may be challenging without detailed instructions or code, possibly limiting its use in broader BIQA applications.
>
> **Response 2:**
> We appreciate the reviewer's concern about implementation complexity. To facilitate the adoption and reproduction of our method, **we will make our complete source code and all associated images publicly available.**

---

> ### Author Response · Authors · 2024-11-27
> **Looking forward to your feedback**
>
> Dear Reviewer aBSv,
>
> Thank you for your valuable and constructive feedback, which has greatly helped improve our work.
>
> With the paper revision deadline approaching, we would be grateful if you could review our response and let us know if our explanations and revisions have adequately addressed your concerns. Any further comments or suggestions you may have would be greatly valued.
>
> We sincerely appreciate your time and expertise and look forward to your feedback.
>
> Best regards,
> The Authors

---

> > ### Comment · Reviewer_aBSv · 2024-11-27
> >
> > Thank you for your response. You have already answered my question. I correspondingly increased my score.

---

### Official Review · Reviewer_KVeG · 2024-11-07

**Soundness:** 2
**Presentation:** 3
**Contribution:** 1
**Rating:** 5
**Confidence:** 4

**Summary:**

The paper proposes a framework, SynDR-IQA, for improving blind image quality assessment (BIQA) models’ generalization by reshaping synthetic data distributions. It introduces two core strategies—distribution-aware diverse content upsampling (DDCUp) and density-aware redundant cluster downsampling (DRCDown)—to balance synthetic data diversity and redundancy, improving models' performance across synthetic and authentic datasets.

**Strengths:**

1. Identifies and addresses critical limitations in synthetic data for BIQA, specifically clustered feature distributions that hinder generalization.
2. Develops SynDR-IQA, which reshapes synthetic data distribution through theoretically grounded strategies (DDCUp and DRCDown), promoting model generalizability.
3. Extensive experiments confirm SynDR-IQA’s superior performance over existing methods in multiple cross-dataset scenarios.

**Weaknesses:**

See the questions

**Questions:**

1. The experimental setup and Table I could be improved by using a larger synthetic dataset annotated with human or pseudo labels, similar to the annotation method used in RankIQA. This would more effectively showcase the advantages of learning from synthetic data for BIQA tasks.


2. The paper’s selection of UDA (unsupervised domain adaptation) methods lacks classical approaches. Including comparisons with well-established UDA methods, such as Maximum Mean Discrepancy (MMD), Domain-Adversarial Neural Networks (DANN), and optimal transport for domain adaptation, would provide a more comprehensive analysis of SynDR-IQA’s performance.


3. There are numerous typographical errors throughout the paper, including "Kadid-10k" on line 521, and inconsistencies in capitalization on lines 281 to 283. Careful proofreading is needed to correct these and other similar errors.

---

> ### Author Response · Authors · 2024-11-22
> **Response to Reviewer KVeG (Part1/2)**
>
> **Question 1:** The experimental setup and Table I could be improved by using a larger synthetic dataset annotated with human or pseudo labels, similar to the annotation method used in RankIQA. This would more effectively showcase the advantages of learning from synthetic data for BIQA tasks.
>
> **Response 1:**
> We appreciate this insightful comment. However, we would like to clarify several points about our experimental setup and contribution.
>
> **Dataset Selection**: We chose KADID-10k as it is the **largest** human-annotated synthetic distortion dataset among those commonly used for synthetic generalization experiments. This choice allows for fair comparison with existing methods.
>
> **Why not use RankIQA?** While RankIQA's pseudo-label approach can improve performance on synthetic datasets, **its effectiveness on real-world distortions is limited**. As shown in Table I, our method significantly outperforms RankIQA across all test datasets.
>
> **Our contribution**:
> 1. We **reveal a key phenomenon**: models trained on synthetic data develop discrete clustered feature distributions, which fundamentally limits their generalization capability to real-world scenarios.
> 2. Through theoretical derivation, we analyzed how sample diversity and redundancy impact model generalization error, **establishing a theoretical foundation** for improving model's generalization capability.
> 3. Based on this theoretical analysis, we proposed the SynDR-IQA framework reshaping synthetic data distribution to enhancing the generalization capability of BIQA models. **Our method demonstrates significant effectiveness in synthetic-to-real generalization.** Beyond the results presented in the paper, we further compare our model (trained on KADID-10k) with state-of-the-art models trained on SPAQ (an authentic distortion dataset with 11k images) when testing on LIVEC and KonIQ-10k. While there is still a gap compared to models trained on authentic distortions, **our method achieves competitive performance despite using only synthetic data for training**.
>
> |                 |                  | LIVEC       | KonIQ-10k   | Average     |
> | --------------- | ---------------- | ----------- | ----------- | ----------- |
> | Methods         | Training Dataset | SRCC PLCC   | SRCC PLCC   | SRCC PLCC   |
> | DBCNN           | SPAQ             | 0.702 0.748 | 0.684 0.702 | 0.693 0.725 |
> | MUSIQ           | SPAQ             | 0.813 0.798 | 0.753 0.680 | 0.783 0.739 |
> | CLIPIQA+        | SPAQ             | 0.719 0.755 | 0.753 0.777 | 0.736 0.766 |
> | SynDR-IQA(ours) | KADID-10k        | 0.713 0.714 | 0.727 0.735 | 0.720 0.725 |
>
> Moreover, our work has broader implications for real-world IQA research:
> 1. Theoretical Insights: Our theoretical analysis of how data diversity and redundancy affect model generalization provides valuable **guidance for future dataset construction and method development** in both synthetic and real-world scenarios.
> 2. Joint Learning: Previous studies have shown that incorporating synthetic distortion data can enhance performance in joint learning settings with real-world data [1]. Our improved synthetic data distribution could potentially further boost performance in such scenarios.
> 3. Incremental Learning: Our proposed DDCUp and DRCDown strategies offer valuable insights into incremental learning methods, particularly on how to maintain distribution consistency and optimize data sampling.
>
> These aspects demonstrate that our work, while focused on synthetic-to-real transfer, has significant implications for advancing real-world IQA research more broadly.
>
> [1] Zhang W, Ma K, Zhai G, et al. Uncertainty-aware blind image quality assessment in the laboratory and wild[J]. IEEE Transactions on Image Processing, 2021, 30: 3474-3486.

---

> ### Author Response · Authors · 2024-11-22
> **Response to Reviewer KVeG (Part2/2)**
>
> **Question 2:** The paper’s selection of UDA (unsupervised domain adaptation) methods lacks classical approaches. Including comparisons with well-established UDA methods, such as Maximum Mean Discrepancy (MMD), Domain-Adversarial Neural Networks (DANN), and optimal transport for domain adaptation, would provide a more comprehensive analysis of SynDR-IQA’s performance.
>
> **Response 2:**
> We appreciate the reviewer's suggestion about including classical UDA methods in our comparisons. Our current comparison already includes RankDA, an **IQA-specific improvement over MMD**, which demonstrated through their ablation study (TABLE VII in [1]) that basic MMD has limited effectiveness for IQA tasks. **Following the reviewer's suggestion, we have further included DANN in our comparison. The results has been added in Table 1.** The experimental results are shown below:
>
> |                 | LIVEC               | KonIQ-10k           | BID                 | Average             |
> | --------------- | ------------------- | ------------------- | ------------------- | ------------------- |
> | Methods         | SRCC PLCC           | SRCC PLCC           | SRCC PLCC           | SRCC PLCC           |
> | DANN            | 0.499 0.484         | 0.638 0.636         | 0.586 0.510         | 0.574 0.543         |
> | RankDA          | 0.451 0.455         | 0.638 0.623         | 0.535 0.582         | 0.542 0.553         |
> | SynDR-IQA(ours) | **0.713** **0.714** | **0.727** **0.735** | **0.788** **0.764** | **0.743** **0.737** |
>
> The results clearly demonstrate the superiority of our approach, with **significant improvements of 16.9% and 20.1% in average SRCC** over DANN and RankDA respectively. While these classical UDA methods provide valuable baselines, **our approach more effectively addresses the unique challenges of synthetic-to-real generalization in IQA tasks**.
>
> [1] Chen B, Li H, Fan H, et al. No-reference screen content image quality assessment with unsupervised domain adaptation[J]. IEEE Transactions on Image Processing, 2021, 30: 5463-5476.
>
> ---
> **Question 3:** There are numerous typographical errors throughout the paper, including "Kadid-10k" on line 521, and inconsistencies in capitalization on lines 281 to 283. Careful proofreading is needed to correct these and other similar errors.
>
> **Response 3:**
> We sincerely thank the reviewers for their careful attention to detail in identifying these typographical errors. We have conducted a comprehensive review of the manuscript and have proofread all noted issues to the best of our ability.

---

> ### Author Response · Authors · 2024-11-27
> **Looking forward to your feedback**
>
> Dear Reviewer KVeG,
>
> Thank you for your valuable and constructive feedback, which has greatly helped improve our work.
>
> With the paper revision deadline approaching, we would be grateful if you could review our response and let us know if our explanations and revisions have adequately addressed your concerns. Any further comments or suggestions you may have would be greatly valued.
>
> We sincerely appreciate your time and expertise and look forward to your feedback.
>
> Best regards,
> The Authors

---

> ### Author Response · Authors · 2024-12-02
> **Looking forward to your feedback**
>
> Dear Reviewer KVeG,
>
> We sincerely thank you for your thorough review and valuable feedback. As the discussion phase is coming to a close (**with less than 2 days remaining for reviewer responses**), we would greatly appreciate it if you could review our responses and share any additional thoughts.
>
> We believe we have addressed all the key points raised in your initial review, and we welcome the opportunity to clarify any remaining concerns to ensure a fully informed assessment.
>
> Thank you again for your time and dedication.
>
> Best regards,
> The Authors

---

> ### Comment · Reviewer_KVeG · 2024-12-02
>
> The feedback addresses part of my concern but does not fully convince me.

---

### Meta-Review · Area_Chair_EW1M · 2024-12-20

**Metareview:**

The paper presents SynDR-IQA to enhance BIQA model generalization by reshaping synthetic data distributions.  It identifies clustered feature distributions in synthetic data and proposes two strategies.  Experiments across multiple cross-dataset settings show the method's effectiveness. Strengths include the innovative SynDR-IQA framework, theoretical analysis foundation, extensive experimental validations, and integration with existing models.  However, weaknesses exist.  Failure case analysis and real-world data applicability discussion could be enhanced.  The complexity of the proposed strategies may pose practical challenges, and more details are needed. Overall, the paper makes a valuable contribution to BIQA.  The proposed method has potential, but revisions are required.  The authors addressed some reviewer concerns, yet further clarification in certain areas would strengthen the paper.  Thus, this paper is slightly below the acceptance threshold.

**Additional Comments On Reviewer Discussion:**

Reviewers raised various concerns.  KVeG suggested a larger synthetic dataset and more UDA method comparisons.  The authors justified dataset choice and added DANN, but the reviewer remained unconvinced.  aBSv pointed out the lack of IQA result images and other issues.  The authors explained and provided visual examples, satisfying the reviewer.  zhhN questioned the method's complexity and generalization explanations.  The authors clarified and argued their focus, though the reviewer wanted more experimental illustrations.  kTZx had multiple concerns, and the authors provided detailed responses, mostly addressing the reviewer's concerns.  Gos6 asked for better motivation elaboration and other improvements.  The authors responded, partially addressing the concerns. In general, the authors were responsive, but some issues need more attention.  More experimental evidence and enhanced discussion on limitations and applicability would be beneficial for acceptance.

---

### Decision · Program_Chairs · 2025-01-22

Reject